# Finite Sample Analysis of Linear Temporal Difference Learning with Arbitrary Features

**Zixuan Xie**[*]
University of Virginia
xie.zixuan@email.virginia.edu

**Xinyu Liu**[*]
University of Virginia
xinyuliu@virginia.edu

**Rohan Chandra**
University of Virginia
rohanchandra@virginia.edu

**Shangtong Zhang**
University of Virginia
shangtong@virginia.edu

## Abstract

Linear TD($\lambda$) is one of the most fundamental reinforcement learning algorithms for policy evaluation. Previously, convergence rates are typically established under the assumption of linearly independent features, which does not hold in many practical scenarios. This paper instead establishes the first $L^2$ convergence rates for linear TD($\lambda$) operating under arbitrary features, without making any algorithmic modification or additional assumptions. Our results apply to both the discounted and average-reward settings. To address the potential non-uniqueness of solutions resulting from arbitrary features, we develop a novel stochastic approximation result featuring convergence rates to the solution set instead of a single point.

## 1 Introduction

Temporal difference learning (TD, Sutton [1988]) is a fundamental algorithm in reinforcement learning (RL, Sutton and Barto [2018]), enabling efficient policy evaluation by combining dynamic programming [Bellman, 1966] with stochastic approximation (SA, Benveniste et al. [1990], Kushner and Yin [2003], Borkar [2009]). Its linear variant, linear TD($\lambda$) [Sutton, 1988], emerges as a practical extension, employing linear function approximation to tackle large or continuous state spaces where tabular representations become impractical. Linear TD($\lambda$) takes the dot product between features and weights to compute the approximated value. Establishing theoretical guarantees for linear TD($\lambda$), particularly convergence rates, has been a major focus of research. Most existing works (Table 1), however, require the features used in linear TD to be linearly independent. As argued in Wang and Zhang [2024], this assumption is impractical in many scenarios. For example, in continual learning with sequentially arriving data [Ring, 1994, Khetarpal et al., 2022, Abel et al., 2023], there is no way to rigorously verify whether the features are independent or not. See Wang and Zhang [2024] for more discussion on the restrictions of the feature independence assumptions. Furthermore, Dayan [1992], Tsitsiklis and Roy [1996, 1999] also outline the elimination of the linear independence assumption as a future research direction.

While efforts have been made to eliminate the linear independence assumption [Wang and Zhang, 2024], they only provide asymptotic (almost sure) convergence guarantees in the discounted setting. By contrast, this paper establishes **the first $L^2$ convergence rates for linear TD($\lambda$) with arbitrary features in both discounted and average-reward settings**. This success is enabled by a novel stochastic approximation result (Theorem 3) concerning the convergence rates to a solution set instead of a single point, driven by a novel Lyapunov function. This new result provides a unified approach

---

[*]Equal contribution

applicable to both discounted (Theorem 1) and average-reward (Theorem 2) settings. Notably, we do not make any algorithmic modification and do not introduce any additional assumptions. Table 1 provides a detailed comparison of existing theoretical analyses for linear TD($\lambda$), contextualizing our contributions within the landscape of prior work.

| | Setting | Features | Noise Type | Rate |
|---|---|---|---|---|
| Tsitsiklis and Roy [1996] | $\gamma < 1$ | Independent | Markovian | |
| Bhandari et al. [2018] | $\gamma < 1$ | Independent | Markovian | ✓ |
| Lakshminarayanan and Szepesvári [2018] | $\gamma < 1$ | Independent | i.i.d. | ✓ |
| Srikant and Ying [2019] | $\gamma < 1$ | Independent | Markovian | ✓ |
| Wang and Zhang [2024] | $\gamma < 1$ | Arbitrary | Markovian | |
| Chen et al. [2025a] | $\gamma < 1$ | Independent | i.i.d. | ✓ |
| Mitra [2025] | $\gamma < 1$ | Independent | Markovian | ✓ |
| Theorem 1 | $\gamma < 1$ | Arbitrary | Markovian | ✓ |
| Tsitsiklis and Roy [1999] | $\gamma = 1$ | Independent | Markovian | |
| Zhang et al. [2021c] | $\gamma = 1$ | Independent | Markovian | ✓ |
| Chen et al. [2025b] | $\gamma = 1$ | Independent | Markovian | ✓ |
| Theorem 2 | $\gamma = 1$ | Arbitrary | Markovian | ✓ |

Table 1: Comparison of finite-sample analyses for linear TD($\lambda$). "Setting" indicates the problem setting: $\gamma < 1$ stands for the discounted setting and $\gamma = 1$ stands for the average reward setting. "Features" describes assumptions on the features. "Independent" indicates linear independence is assumed. "Arbitrary" indicates no assumption is made on features. "Noise Type" indicates the data generation process: Markovian samples or independent and identically distributed (i.i.d.) samples. "Rate" is checked if a convergence rate is provided.

## 2 Background

**Notations.** We use $\langle x, y \rangle \doteq x^\top y$ to denote the standard inner product in Euclidean spaces and $\|\cdot\|$ to denote the $\ell_2$ norm for vectors and the associated induced operator norm (i.e., the spectral norm) for matrices, unless stated otherwise. A function $f$ is said to be $L$-smooth (w.r.t. $\|\cdot\|$) if $\forall w, w'$, $f(w') \leq f(w) + \langle \nabla f(w), w' - w \rangle + \frac{L}{2}\|w' - w\|^2$. For a matrix $A$, $\mathrm{col}(A)$ denotes its column space, $\ker(A)$ denotes its kernel, and $A^\dagger$ denotes its Moore-Penrose inverse. When $x$ is a point and $U$ is a set, we denote $d(x, U) \doteq \inf_{y \in U} \|x - y\|$ as the Euclidean distance from $x$ to $U$. For sets $U, V$, their Minkowski sum is $U + V \doteq \{u + v \mid u \in U, v \in V\}$; and $U^\perp$ denotes the orthogonal complement of $U$. We use $\mathbf{0}$ and $\mathbf{1}$ to denote the zero vector and the all-ones vector respectively, where the dimension is clear from context. For any square matrix $A \in \mathbb{R}^{d \times d}$ (not necessarily symmetric), we say $A$ is negative definite (n.d.) if there exists a $\xi > 0$ such that $x^\top A x \leq -\xi \|x\|^2 \ \forall x \in \mathbb{R}^d$. For any set $E \subseteq \mathbb{R}^d$, we say $A$ is n.d. on $E$ if there exists a $\xi > 0$ such that $x^\top A x \leq -\xi \|x\|^2 \ \forall x \in E$. A is negative semidefinite (n.s.d.) if $\xi = 0$ in the above definition.

**Markov Decision Processes.** We consider an infinite horizon Markov Decision Process (MDP, Bellman [1957]) defined by a tuple $(\mathcal{S}, \mathcal{A}, p, r, p_0)$, where $\mathcal{S}$ is a finite set of states, $\mathcal{A}$ is a finite set of actions, $p : \mathcal{S} \times \mathcal{S} \times \mathcal{A} \to [0, 1]$ is the transition probability function, $r : \mathcal{S} \times \mathcal{A} \to \mathbb{R}$ is the reward function, and $p_0 : \mathcal{S} \to [0, 1]$ denotes the initial distribution. In this paper, we focus on the policy evaluation problem, where the goal is to estimate the value function of an arbitrary policy $\pi : \mathcal{A} \times \mathcal{S} \to [0, 1]$. At the time step 0, an initial state $S_0$ is sampled from $p_0$. At each subsequent time step $t$, the agent observes state $S_t \in \mathcal{S}$, executes an action $A_t \sim \pi(\cdot|S_t)$, receives reward $R_{t+1} \doteq r(S_t, A_t)$, and transitions to the next state $S_{t+1} \sim p(\cdot|S_t, A_t)$. We use $P_\pi$ to denote the state transition matrix induced by the policy $\pi$, i.e., $P_\pi[s, s'] = \sum_{a \in \mathcal{A}} \pi(a|s)p(s'|s, a)$. Let $d_\pi \in \mathbb{R}^{|\mathcal{S}|}$ be the stationary distribution of the Markov chain induced by the policy $\pi$. We use $D_\pi$ to denote the diagonal matrix whose diagonal is $d_\pi$.

**Linear Function Approximation.** In this paper, we use linear function approximation to approximate value functions $v_\pi : \mathcal{S} \to \mathbb{R}$ (to be defined shortly). We consider a feature mapping $x : \mathcal{S} \to \mathbb{R}^d$ and a weight vector $w \in \mathbb{R}^d$. We then approximate $v_\pi(s)$ with $x(s)^\top w$. We use $X \in \mathbb{R}^{|\mathcal{S}| \times d}$ to denote

the feature matrix, where the $s$-th row of $X$ is $x(s)^\top$. The approximated state-value function across all states can then be represented as the vector $Xw \in \mathbb{R}^{|\mathcal{S}|}$. The goal is thus to find a $w$ such that $Xw$ closely approximates $v_\pi$.

**Discounted Setting.** In the discounted setting, we introduce a discount factor $\gamma \in [0, 1)$. The (discounted) value function $v_\pi : \mathcal{S} \to \mathbb{R}$ for policy $\pi$ is defined as $v_\pi(s) \doteq \mathbb{E}\left[\sum_{i=0}^\infty \gamma^i R_{t+i+1} \big| S_t = s\right]$. We define the Bellman operator $\mathcal{T} : \mathbb{R}^{|\mathcal{S}|} \to \mathbb{R}^{|\mathcal{S}|}$ as $\mathcal{T}v \doteq r_\pi + \gamma P_\pi v$, where $r_\pi \in \mathbb{R}^{|\mathcal{S}|}$ is the vector of expected immediate rewards under $\pi$, with components $r_\pi(s) = \sum_a \pi(a|s)r(s, a)$. With a $\lambda \in [0, 1]$, the $\lambda$-weighted Bellman operator $\mathcal{T}_\lambda$ is defined as $\mathcal{T}_\lambda v \doteq (1 - \lambda)\sum_{m=0}^\infty \lambda^m \mathcal{T}^{m+1}v = r_\lambda + \gamma P_\lambda v$, where

$$r_\lambda = \sum_{k=0}^\infty (\lambda\gamma)^k P_\pi^k r_\pi = (1 - \gamma\lambda P_\pi)^{-1} r_\pi,$$
$$P_\lambda = (1 - \lambda)\sum_{m=0}^\infty (\lambda\gamma)^m P_\pi^{m+1} = (1 - \lambda)(1 - \gamma\lambda P_\pi)^{-1} P_\pi.$$

This represents a weighted average of multi-step applications of $\mathcal{T}$. It is well-known that $v_\pi$ is the unique fixed point of $\mathcal{T}_\lambda$ [Bertsekas and Tsitsiklis, 1996]. Linear TD($\lambda$) is a family of TD learning algorithms that use eligibility traces to estimate $v_\pi(s)$ of the fixed policy $\pi$ with linear function approximation. The algorithm maintains a weight vector $w_t \in \mathbb{R}^d$ and an eligibility trace vector $e_t \in \mathbb{R}^d$, with the following update rules:

$$w_{t+1} = w_t + \alpha_t(R_{t+1} + \gamma x(S_{t+1})^\top w_t - x(S_t)^\top w_t)e_t,$$
$$e_t = \gamma\lambda e_{t-1} + x(S_t), \ e_{-1} = \mathbf{0}. \qquad \text{(Discounted TD)}$$

Here, $\{\alpha_t\}$ is the learning rate. The eligibility trace $e_t$ tracks recently visited states, assigning credit for the prediction error to multiple preceding states. Let

$$A \doteq X^\top D_\pi(\gamma P_\lambda - I)X, \ b \doteq X^\top D_\pi r_\lambda, \ W_* \doteq \{w|Aw + b = \mathbf{0}\}.$$

If $X$ has a full column rank, Tsitsiklis and Roy [1996] proves that $W_*$ is a singleton and $\{w_t\}$ converge to $-A^{-1}b$ almost surely. A key result used by Tsitsiklis and Roy [1996] is that the matrix $D_\pi(\gamma P_\lambda - I)$ is n.d. [Sutton, 1988]. As a result, the $A$ matrix is also n.d. when $X$ has a full column rank. Wang and Zhang [2024] prove, without making any assumption on $X$, that $W_*$ is always nonempty and the $\{w_t\}$ converges to $W_*$ almost surely. A key challenge there is that without making assumptions on $X$, $A$ is only n.s.d.

**Average-Reward Setting.** In the average-reward setting, the overall performance of a policy $\pi$ is measured by the average reward $J_\pi \doteq \lim_{T\to\infty} \frac{1}{T}\mathbb{E}\left[\sum_{t=0}^{T-1} R_t\right]$. The corresponding (differential) value function is defined as $\overline{v}_\pi(s) = \lim_{T\to\infty} \frac{1}{T}\sum_{i=0}^{T-1} \mathbb{E}[(r(S_{t+i}, A_{t+i}) - J_\pi)|S_t = s]$. We define the Bellman operator $\overline{\mathcal{T}} : \mathbb{R}^{|\mathcal{S}|} \to \mathbb{R}^{|\mathcal{S}|}$ as $\overline{\mathcal{T}}v \doteq r_\pi - J_\pi\mathbf{1} + P_\pi v$. Similarly, the $\lambda$-weighted counterpart $\overline{\mathcal{T}}_\lambda$ is defined as $\overline{\mathcal{T}}_\lambda v \doteq r_\lambda - \frac{J_\pi}{1-\lambda}\mathbf{1} + P_\lambda v$. Although $\overline{v}_\pi$ is a fixed point of $\overline{\mathcal{T}}_\lambda$, it is not the unique fixed point. In fact,

$$\{\overline{v}_\pi + c\mathbf{1} \mid c \in \mathbb{R}\} \qquad (2)$$

are all the fixed points of $\overline{\mathcal{T}}_\lambda$ [Puterman, 2014]. Linear average-reward TD($\lambda$) is an algorithm for estimating both $J_\pi$ and $\overline{v}_\pi$ using linear function approximation and eligibility traces. The update rules are

$$e_t = \lambda e_{t-1} + x(S_t), \ e_{-1} = \mathbf{0},$$
$$w_{t+1} = w_t + \alpha_t(R_{t+1} - \hat{J}_t + x(S_{t+1})^\top w_t - x(S_t)^\top w_t)e_t,$$
$$\hat{J}_{t+1} = \hat{J}_t + \beta_t(R_{t+1} - \hat{J}_t), \qquad \text{(Average Reward TD)}$$

where $\{\alpha_t\}$ and $\{\beta_t\}$ are learning rates. Let

$$\overline{A} \doteq X^\top D_\pi(P_\lambda - I)X, \ \overline{b} \doteq X^\top D_\pi(r_\lambda - \frac{J_\pi}{1-\lambda}\mathbf{1}), \ \overline{W}_* \doteq \{w|\overline{A}w + \overline{b} = \mathbf{0}\}. \qquad (4)$$

If $X$ has a full column rank and $\mathbf{1} \notin \mathrm{col}(X)$, Tsitsiklis and Roy [1999] proves that $\overline{W}_*$ is a singleton and $\{w_t\}$ converge to $-\overline{A}^{-1}\overline{b}$ almost surely. This is made possible by an important fact from the Perron-Frobenius theorem (see, e.g., Seneta [2006]) that

$$\{w|w^\top D_\pi(P_\lambda - I)w = 0\} = \{c\mathbf{1}|c \in \mathbb{R}\}. \qquad (5)$$

Zhang et al. [2021c] further provides a convergence rate, still assuming $X$ has a full column rank but without assuming $\mathbf{1} \notin \mathrm{col}(X)$. When $X$ does not have a full column rank, to our knowledge, it is even not clear whether $\overline{W}_*$ is always nonempty or not, much less the behavior of $\{w_t\}$.

# 3 Main Results

We start with our assumptions. As promised, we do not make any assumption on $X$.

**Assumption 3.1.** The Markov chain associated with $P_\pi$ is irreducible and aperiodic.

**Assumption LR.** *The learning rates are $\alpha_t = \frac{\alpha}{(t+t_0)^\xi}$ and $\beta_t = c_\beta \alpha_t$, where $\xi \in (0.5, 1]$, $\alpha > 0$, $t_0 > 0$, and $c_\beta > 0$ are constants.*

**Discounted Setting.** Wang and Zhang [2024] proves the almost sure convergence of (Discounted TD) with arbitrary features by using $\|w - w_*\|^2$ with an arbitrary and fixed $w_* \in W_*$ as a Lyapunov function and analyzing the property of the ODE $\frac{dw(t)}{dt} = Aw(t)$. Since $A$ is only n.s.d., Wang and Zhang [2024] conducts their analysis in the complex number field. In this work, instead of following the ODE-based analysis originating from Tsitsiklis and Roy [1996], Borkar and Meyn [2000], we extend Srikant and Ying [2019] to obtain convergence rates by using $d(w, W_*)^2$ as the Lyapunov function. To our knowledge, this is the first time that such distance function to a set is used as the Lyapunov function to analyze RL algorithms, which is our key technical contribution from the methodology aspect. According to Theorem 1 of Wang and Zhang [2024], $W_*$ is nonempty, and apparently convex and closed.[2] Let $\Gamma(w) \doteq \arg\min_{w_* \in W_*} \|w - w_*\|$ be the orthogonal projection to $W_*$. We then define $L(w) \doteq \frac{1}{2} d(w, W_*)^2 = \frac{1}{2} \|w - \Gamma(w)\|^2$. Two important and highly non-trivial observations are

(i) $\nabla L(w) = w - \Gamma(w)$ (Example 3.31 of Beck [2017]),

(ii) $L(w)$ is 1-smooth w.r.t. $\|\cdot\|$ (Example 5.5 of Beck [2017]).

Both (i) and (ii) result from the fact that $W_*$ is nonempty, closed, and convex. Using $L(w)$ as the Lyapunov function together with more characterization of $\nabla L(w)$ (Section 5.2), we obtain

**Theorem 1.** *Let Assumptions 3.1 and LR hold and $\lambda \in [0, 1]$. Then for sufficiently large $t_0$ and $\alpha$, there exist some constants $C_{Thm1}$ and $\kappa_1 \doteq \alpha C_7 > 1$ such that the iterates $\{w_t\}$ generated by (Discounted TD) satisfy for all $t$*

$$\mathbb{E}\big[d(w_t, W_*)^2\big] \leq C_{Thm1}\Big(\Big(\tfrac{t_0}{t}\Big)^{\lfloor \kappa_1 \rfloor} d(w_0, w_*)^2 + \Big(\tfrac{\ln(t+t_0)}{(t+t_0)^{\min(2\xi-1, \lfloor \kappa_1 \rfloor - 1)}}\Big)\Big).$$

The proof is in Section 5.2. Notably, Lemma 3 of Wang and Zhang [2024] states that for any $w_*, w_{**} \in W_*$, it holds that $Xw_* = Xw_{**}$. We then define

$$\hat{v}_\pi \doteq Xw_* \tag{6}$$

for any $w_* \in W_*$. Theorem 1 then also gives the $L^2$ convergence rate of the value estimate, i.e., the rate at which $Xw_t$ converges to $\hat{v}_\pi$. The value estimate $\hat{v}_\pi$ is the unique fixed point of a projected Bellman equation. See Wang and Zhang [2024] for more discussion on the property of $\hat{v}_\pi$. Additionally, by choosing a sufficiently large $\alpha$, we can ensure $\lfloor \kappa_1 \rfloor - 1 \geq 2\xi - 1$, so the rate is determined by the exponent $2\xi - 1$. For the standard choice $\xi = 1$, the resulting rate becomes $\mathcal{O}(\ln t/t)$, which matches existing analyses that assume linearly independent features [Bhandari et al., 2018, Srikant and Ying, 2019]. An analogous observation holds for Theorems 2 and 3 as well, since their corresponding $\kappa$ is also proportional to $\alpha$.

**Average Reward Setting.** Characterizing $\overline{W}_*$ is much more challenging. We first present a novel decomposition of the feature matrix $X$. To this end, define $m \doteq \text{rank}(X) \leq \min\{|\mathcal{S}|, d\}$. If $m = 0$, all the results in this work are trivial and we thus discuss only the case $m \geq 1$.

**Lemma 1.** *There exist matrices $X_1, X_2$ such that $X = X_1 + X_2$ with the following properties (1) $\text{rank}(X_1) = m - \mathbb{I}_{\mathbf{1} \in \text{col}(X)}$ and $\mathbf{1} \notin X_1$ (2) $X_2 = \mathbf{1}\theta^\top$ with $\theta \in \mathbb{R}^d$.*

The proof is in Section B.1 with $\mathbb{I}$ being the indicator function. Essentially, $X_2$ is a rank one matrix with identical rows $\theta$ (i.e., the $i$-th column of $X_2$ is $\theta_i \mathbf{1}$). To our knowledge, this is the first time that such decomposition is used to analyze average-reward RL algorithms, which is our second

---

[2]This theorem only discusses the case of $\lambda = 0$. The proof for a general $\lambda \in [0, 1]$ is exactly the same up to change of notations.

technical contribution from the methodology aspect. This decomposition is useful in three aspects. First, we have $\overline{A} = X_1^\top D_\pi (P_\lambda - I) X_1$ (Lemma 14). Second, this decomposition is the key to prove that $\overline{W}_*$ is nonempty (Lemma 15). Third, this decomposition is the key to characterize $\overline{W}_*$ in that $\overline{W}_* = \{\overline{w}_*\} + \ker(X_1)$ with $\overline{w}_*$ being any vector in $\overline{W}_*$ (Lemma 16). To better understand this characterization, we note that $\ker(X_1) = \{w | Xw = c\mathbf{1}, c \in \mathbb{R}\}$ (Lemma 16). As a result, adding any $w_0 \in \ker(X_1)$ to a weight vector $w$ changes the resulting value function $Xw$ only by $c\mathbf{1}$. Two values $v_1$ and $v_2$ can be considered "duplication" if $v_1 - v_2 = c\mathbf{1}$ (cf. (2)). So intuitively, $\ker(X_1)$ is the source of the "duplication". With the help of this novel decomposition, we obtain

**Theorem 2.** *Let Assumptions 3.1 and LR hold and $\lambda \in [0, 1)$. Then for sufficiently large $\alpha$, $t_0$ and $c_\beta$, there exist some constants $C_{Thm2}$ and $\kappa_2 \doteq \alpha C_{10} > 1$ such that the iterates $\{w_t\}$ generated by (Average Reward TD) satisfy for all $t$*

$$\mathbb{E}\Big[(\hat{J}_t - J_\pi)^2 + d(w_t, \overline{W}_*)^2\Big] \leq C_{Thm2}\big(\tfrac{t_0}{t}\big)^{\lfloor \kappa_2 \rfloor} \Big[(\hat{J}_0 - J_\pi)^2 + d(w_0, \overline{W}_*)^2\Big]$$
$$+ C_{Thm2}\Big(\tfrac{\ln(t+t_0)}{(t+t_0)^{\min(2\xi-1, \lfloor \kappa_2 \rfloor -1)}}\Big).$$

The proof is in Section 5.3.

**Stochastic Approximation.** We now present a general stochastic approximation result to prove Theorems 1 and 2. The notations in this part are independent of the rest of the paper. We consider a general iterative update rule for a weight vector $w \in \mathbb{R}^d$, driven by a time-homogeneous Markov chain $\{Y_t\}$ evolving in a possibly infinite space $\mathcal{Y}$:

$$w_{t+1} = w_t + \alpha_t H(w_t, Y_{t+1}), \tag{SA}$$

where $H : \mathbb{R}^d \times \mathcal{Y} \to \mathbb{R}^d$ defines the incremental update.

**Assumption A1.** *There exists a constant $C_{A1}$ such that $\sup_{y \in \mathcal{Y}} \|H(0, y)\| < \infty$,*

$$\|H(w_1, y) - H(w_2, y)\| \leq C_{A1} \|w_1 - w_2\| \quad \forall w_1, w_2, y.$$

**Assumption A2.** *$\{Y_t\}$ has a unique stationary distribution $d_{\mathcal{Y}}$.*

Let $h(w) \doteq \mathbb{E}_{y \sim d_{\mathcal{Y}}}[H(w, y)]$. Assumption A1 then immediately implies that

$$\|h(w_1) - h(w_2)\| \leq C_{A1} \|w_1 - w_2\| \quad \forall w_1, w_2.$$

In many existing works about stochastic approximation [Borkar and Meyn, 2000, Chen et al., 2023b, Qian et al., 2024, Borkar et al., 2025], it is assumed that $h(w) = 0$ adopts a unique solution. To work with the challenges of linear TD with arbitrary features, we relax this assumption and consider a set $W_*$. Importantly, $W_*$ does not need to contain all solutions to $h(w) = 0$. Instead, we make the following assumptions on $W_*$.

**Assumption A3.** *$W_*$ is nonempty, closed, and convex.*

Notably, $W_*$ does not need to be bounded. Assumption A3 ensures that the orthogonal projection to $W_*$ is well defined, allowing us to define $\Gamma(w) \doteq \arg\min_{w_* \in W_*} \|w - w_*\|$, $L(w) \doteq \frac{1}{2}\|w - \Gamma(w)\|^2$. As discussed before, Assumption A3 ensures that $\nabla L(w) = w - \Gamma(w)$ and $L$ is 1-smooth w.r.t. $\|\cdot\|$ [Beck, 2017]. We further assume that the expected update $h(w_t)$ decreases $L(w_t)$ in the following sense, making $L(w)$ a candidate Lyapunov function.

**Assumption A4.** *There exists a constant $C_{A4} > 0$ such that almost surely,*

$$\langle \nabla L(w_t), h(w_t) \rangle \leq -C_{A4} L(w_t).$$

Lastly, we make the most "unnatural" assumption of $W_*$.

**Assumption A5.** *There exists a matrix $X$ and constants $C_{A5}$ and $\tau \in [0, 1)$ such that (1) $\forall w_* \in W_*$, $\|Xw_*\| \leq C_{A5}$; (2) $\forall w, y$, $\|H(w, y)\| \leq C_{A5}(\|Xw\| + 1)$; (3) For any $n \geq 1$:*

$$\|h(w) - \mathbb{E}[H(w, Y_{t+n})|Y_t]\| \leq C_{A5} \tau^n (\|Xw\| + 1) \tag{7}$$

This assumption is technically motivated but trivially holds in our analyses of (Discounted TD) and (Average Reward TD). Specifically, Assumption A1 immediately leads to at-most-linear growth

$\|H(w, y)\| \leq C_{A1,1}(\|w\| + 1)$ for some constant $C_{A1,1}$. However, this bound is insufficient for our analysis because $\|w\| \leq \|w - \Gamma(w)\| + \|\Gamma(w)\|$ but $\Gamma(w) \in W_*$ can be unbounded. By Assumption A5, we can have $\|Xw\| \leq \|Xw - X\Gamma(w)\| + \|X\Gamma(w)\| \leq \|X\|\|w - \Gamma(w)\| + C_{A5}$. The inequality (7) is related to geometrical mixing of the chain and we additionally include $Xw$ in the bound for the same reason. We now present our general results regarding the convergence rate of (SA) to $W_*$.

**Theorem 3.** *Let Assumptions A1 - A5 and LR hold. Denote $\kappa \doteq \alpha C_{A4}$, then there exist some constants $t_0$ and $C_{Thm3}$, such that the iterates $\{w_t\}$ generated by (SA) satisfy for all $t$*

$$\mathbb{E}[L(w_t)] \leq C_{Thm3,1}\left(\frac{t_0}{t}\right)^{\lfloor \kappa \rfloor} L(w_0) + C_{Thm3,2}\left(\frac{\ln(t+t_0)}{(t+t_0)^{\min(2\xi-1, \lfloor \kappa \rfloor - 1)}}\right).$$

The proof is in Section 5.1. We remark that once we have the recursion in Lemma 5, our theoretical framework can be readily extended to the constant step-size setting (akin to Chen et al. [2023b]), demonstrating its broad applicability.

# 4   Related Works

Most prior works regarding the convergence of linear TD summarized in Table 1 rely on having linearly independent features. In fact, the reliance on feature independence goes beyond linear TD and exists in almost all previous analyses of RL algorithms with linear function approximation, see, e.g., Sutton et al. [2008, 2009], Maei [2011], Hackman [2012], Bo et al. [2015], Yu [2015, 2016], Zou et al. [2019], Yang et al. [2019], Zhang et al. [2020b], Xu et al. [2020a], Zhang et al. [2020a], Xu et al. [2020b], Wu et al. [2020], Chen et al. [2021], Long et al. [2021], Qiu et al. [2021], Zhang et al. [2021a,b], Xu et al. [2021], Zhang et al. [2022], Zhang and Whiteson [2022], Zhang et al. [2023], Chen et al. [2023a], Nicolò et al. [2024], Yue et al. [2024], Swetha et al. [2024], Liu et al. [2025a], Qian and Zhang [2025], Maity and Mitra [2025], Yang et al. [2025], Chen et al. [2025b], Shaan and Siva [2025], Liu et al. [2025c]. But as argued by Dayan [1992], Tsitsiklis and Roy [1996, 1999], Wang and Zhang [2024], relaxing this assumption is an important research direction. This work can be viewed as an extension of Wang and Zhang [2024], Zhang et al. [2021c]. In terms of (Discounted TD), we extend Wang and Zhang [2024] by proving a finite sample analysis. Though we rely on the characterization of $W_*$ from Wang and Zhang [2024], the techniques we use for finite sample analysis are entirely different from the techniques of Wang and Zhang [2024] for almost sure asymptotic convergence. In terms of (Average Reward TD), we extend Zhang et al. [2021c] by allowing $X$ to be arbitrary. Essentially, key to Zhang et al. [2021c] is their proof that $\overline{A}$ is n.d. on a subspace $E$, assuming $X$ has a full column rank. We extend Zhang et al. [2021c] in that we give a finer and more detailed characterization of the counterparts of their $E$ through the novel decomposition of the features (Lemma 1) and establish the n.d. property under weaker conditions (i.e., without assuming $X$ has a full column rank). Importantly, despite relaxing the feature-independence assumption, our convergence rate remains on par with existing finite-sample results obtained under full-rank features [Bhandari et al., 2018, Srikant and Ying, 2019]. Our improvements are made possible by the novel Lyapunov function $L(w)$ and we argue that this Lyapunov function can be used to analyze many other linear RL algorithms with arbitrary features.

In terms of stochastic approximation, our Theorem 3 is novel in that it allows convergence to a possibly unbounded set. By contrast, most prior works about stochastic approximation study convergence to a point [Borkar and Meyn, 2000, Chen et al., 2020, Zhang et al., 2022, Chen et al., 2023b, Qian et al., 2024, Liu et al., 2025a, Borkar et al., 2025, Chen et al., 2025a]. In the case of convergence to a set, most prior works require the set to be bounded [Kushner and Yin, 2003, Borkar, 2009, Liu et al., 2025a,b]. Only a few prior works allow stochastic approximation to converge to an unbounded set, see, e.g., Bravo and Cominetti [2024], Chen [2025], Blaser and Zhang [2025], which apply to only tabular RL algorithms.

# 5   Proofs of the Main Results

## 5.1   Proof of Theorem 3

*Proof.* From the 1-smoothness of $L(w)$ and (SA), we can get

$$L(w_{t+1}) \leq L(w_t) + \alpha_t \langle w_t - \Gamma(w_t), h(w_t) \rangle$$

$$+ \alpha_t \langle w_t - \Gamma(w_t), H(w_t, Y_t) - h(w_t) \rangle + \tfrac{1}{2}\alpha_t^2 \|H(w_t, Y_t)\|^2. \tag{8}$$

We then bound the RHS one by one. $\langle w - \Gamma(w), h(w) \rangle$ is already bounded in Assumption A4.

**Lemma 2.** *There exists a positive constant $C_2$, such that for any $w$,*
$$\|Xw\| \le C_2(\|w - \Gamma(w)\| + 1).$$

The proof is in Section C.1. With Lemma 2 and Assumption A5, the last term in (8) can be bounded easily.

**Lemma 3.** *There exists a constant $C_3$ such that $\|H(w_t, Y_t)\|^2 \le C_3(\|w_t - \Gamma(w_t)\|^2 + 1)$.*

The proof is in Section C.2. To bound $\langle w_t - \Gamma(w_t), H(w_t, Y_t) - h(w_t) \rangle$, leveraging (7), we define
$$\tau_\alpha \doteq \min\{n \ge 0 \mid C_{A5}\tau^n \le \alpha\} \tag{9}$$

as the number of steps that the Markov chain needs to mix to an accuracy $\alpha$. In addition, we denote a shorthand $\alpha_{t_1, t_2} \doteq \sum_{i=t_1}^{t_2} \alpha_i$. Then with techniques from Srikant and Ying [2019], we obtain

**Lemma 4.** *There exists a constant $C_4$ such that*
$$\mathbb{E}[\langle w_t - \Gamma(w_t), H(w_t, Y_t) - h(w_t) \rangle] \le C_4 \alpha_{t-\tau_{\alpha_t}, t-1}(\|w_t - \Gamma(w_t)\|^2 + 1).$$

The proof is in Section C.3. Plugging all the bounds back to (8), we obtain

**Lemma 5.** *There exists some $D_t = \mathcal{O}(\alpha_t \alpha_{t-\tau_{\alpha_t}, t-1})$, such that*
$$\mathbb{E}[L(w_{t+1})] \le (1 - C_{A4}\alpha_t)\mathbb{E}[L(w_t)] + D_t.$$

The proof is in Section C.4. Recursively applying Lemma 5 then completes the proof of Theorem 3 (See Section C.5 for details). $\qquad \square$

In the following sections, we first map the general update (SA) to (Discounted TD) and (Average Reward TD) by defining $H(w, y)$, $h(w)$, and $L(w)$ properly. Then we bound the remaining term $\langle \nabla L(w_t), h(w_t) \rangle$ to complete the proof.

### 5.2   Proof of Theorem 1

*Proof.* We first rewrite (Discounted TD) in the form of (SA). To this end, we define $Y_{t+1} \doteq (S_t, A_t, S_{t+1}, e_t)$, which evolves in an infinite space $\mathcal{Y} \doteq \mathcal{S} \times \mathcal{A} \times \mathcal{S} \times \{e \mid \|e\| \le C_e\}$ with $C_e \doteq \frac{\max_s \|x(s)\|}{1 - \gamma\lambda}$ being the straightforward bound of $\sup_t \|e_t\|$. We define the incremental update $H : \mathbb{R}^d \times \mathcal{Y} \to \mathbb{R}^d$ as
$$H(w, y) = (r(s, a) + \gamma x(s')^\top w - x(s)^\top w)e, \tag{10}$$

using shorthand $y = (s, a, s', e)$. We now proceed to verifying the assumptions of Theorem 3. Assumption A1 is verified by the following lemma.

**Lemma 6.** *There exists some finite $C_6$ such that*
$$\|H(w_1, y) - H(w_2, y)\| \le C_6 \|w_1 - w_2\| \quad \forall w_1, w_2, y.$$
*Moreover, $\sup_{y \in \mathcal{Y}} \|H(0, y)\| < \infty$.*

The proof is in Section D.1.
For Assumption A2, Theorem 3.2 of Yu [2012] confirms that $\{Y_t\}$ has a unique stationary distribution $d_{\mathcal{Y}}$. Yu [2012] also computes that
$$h(w) \doteq \mathbb{E}_{y \sim d_{\mathcal{Y}}}[H(w, y)] = Aw + b.$$

Assumption A3 trivially holds by the definition of $W_*$.
For Assumption A4, the key observation is that $A\Gamma(w) + b = 0$ always holds because $\Gamma(w) \in W_*$. Then we have $h(w) = Aw + b = (Aw + b) - (A\Gamma(w) + b) = A(w - \Gamma(w))$. Thus the term $\langle \nabla L(w), h(w) \rangle$ can be written as $(w - \Gamma(w))^\top A(w - \Gamma(w))$. We now prove that for whatever $X$, it always holds that $A$ is n.d. on $\ker(A)^\perp$.

**Lemma 7.** *There exists a constant $C_7 > 0$ such that for $\forall w \in \ker(A)^\perp$, $w^\top A w \leq -C_7 \|w\|^2$. Furthermore, for any $w \in \mathbb{R}^d$, it holds that $w - \Gamma(w) \in \ker(A)^\perp$.*

The proof is in Section D.3. We then have

$$\langle w_t - \Gamma(w_t), A(w_t - \Gamma(w_t)) \rangle \leq -C_7 \|w_t - \Gamma(w_t)\|^2,$$

which satisfies Assumption A4.
For Assumption A5, (6) verifies Assumption A5(1). Assumption A5(2) is verified by the following lemma.

**Lemma 8.** *There exists a constant $C_8$ such that for $\forall w, y$, $\|H(w, y)\| \leq C_8(\|Xw\| + 1)$.*

The proof is in Section D.4. Assumption A5(3) is verified following a similar procedure as Lemma 6.7 in Bertsekas and Tsitsiklis [1996] (Lemma 18). Invoking Theorem 3 then completes the proof. $\qquad\square$

## 5.3 Proof of Theorem 2

*Proof.* We recall that in view of Lemma 1, $\ker(X_1)$ creates "duplication" in value estimation. We, therefore, define the projection matrix $\Pi \in \mathbb{R}^{d \times d}$ that projects a vector into the orthogonal complement of $\ker(X_1)$, i.e., $\Pi w \doteq \arg\min_{w' \in \ker(X_1)^\perp} \|w - w'\|$. It can be computed that $\Pi = X_1^\dagger X_1$. We now examine the sequence $\{\Pi w_t\}$ with $\{w_t\}$ being the iterates of (Average Reward TD) and consider the combined parameter vector $\widetilde{w}_t \doteq \begin{bmatrix} \hat{J}_t \\ \Pi w_t \end{bmatrix} \in \mathbb{R}^{1+d}$. The following lemma characterizes the evolution of $\widetilde{w}_t$. Let $Y_t = (S_t, A_t, S_{t+1}, e_t) \in \mathcal{S} \times \mathcal{A} \times \mathcal{S} \times \left\{ e \in \mathbb{R}^d \mid \|e\| \leq \frac{\max_s \|x(s)\|}{1 - \lambda} \right\}$, then

**Lemma 9.** *$\widetilde{w}_{t+1} = \widetilde{w}_t + \alpha_t(\widetilde{A}(Y_t)\widetilde{w}_t + \widetilde{b}(Y_t))$, where we have, with $y = (s, a, s', e)$,*

$$\widetilde{A}(y) = \begin{bmatrix} -c_\beta & \mathbf{0} \\ -\Pi e & \Pi e(x(s')^\top - x(s)^\top) \end{bmatrix}, \widetilde{b}(y) = \begin{bmatrix} c_\beta r(s, a) \\ r(s, a)\Pi e \end{bmatrix}.$$

This view is inspired by Zhang et al. [2021c] and the proof is in Section E.1. We now apply Theorem 3 to $\{\widetilde{w}_t\}$.

The verification of Assumptions A1 and A2 is identical to that in Section 5.2 and is thus omitted.
For Assumption A3, we define $\widetilde{W}_* \doteq \left\{ \begin{bmatrix} J_\pi \\ \Pi w \end{bmatrix} \Big| w \in \overline{W}_* \right\}$. It is apparently nonempty, closed, and convex.
For Assumption A4, we define $\widetilde{A} \doteq \mathbb{E}_{y \sim d_\mathcal{Y}} \left[ \widetilde{A}(y) \right]$ and $\widetilde{b} \doteq \mathbb{E}_{y \sim d_\mathcal{Y}} \left[ \widetilde{b}(y) \right]$ and therefore realize the $h$ in (SA) as $h(\widetilde{w}) = \widetilde{A}\widetilde{w} + \widetilde{b}$. Noticing that $\widetilde{A}\Gamma(\widetilde{w}) + \widetilde{b} = \mathbf{0}$ (Lemma 19), we then have $h(\widetilde{w}) = \widetilde{A}(\widetilde{w} - \Gamma(\widetilde{w}))$. The term $\langle \nabla L(\widetilde{w}), h(\widetilde{w}) \rangle$ can thus be written as $(\widetilde{w} - \Gamma(\widetilde{w}))^\top \widetilde{A}(\widetilde{w} - \Gamma(\widetilde{w}))$. Next, we prove that when $c_\beta$ is large enough, $\widetilde{A}$ is n.d. on $\mathbb{R} \times \ker(X_1)^\perp$.

**Lemma 10.** *Let $c_\beta$ be sufficiently large. Then there exists a constant $C_{10} > 0$ such that $\forall z \in \mathbb{R} \times \ker(X_1)^\perp$, $z^T \widetilde{A} z \leq -C_{10}\|z\|^2$.*

The proof is in Section E.3. By definition, we have $\widetilde{w}_t \in \mathbb{R} \times \ker(X_1)^\perp$ and $\Gamma(\widetilde{w}) \in \mathbb{R} \times \ker(X_1)^\perp$. So $\widetilde{w} - \Gamma(\widetilde{w}) \in \mathbb{R} \times \ker(X_1)^\perp$, yielding

$$\langle \widetilde{w}_t - \Gamma(\widetilde{w}_t), \widetilde{A}(\widetilde{w}_t - \Gamma(\widetilde{w}_t)) \rangle \leq -C_{10}\|\widetilde{w}_t - \Gamma(\widetilde{w}_t)\|^2,$$

which verifies Assumption A4.
For Assumption A5, we define $\widetilde{X} = \begin{bmatrix} 1 & \mathbf{0}^\top \\ \mathbf{0} & X \end{bmatrix}$. Assumption A5(1) is verified below.

**Lemma 11.** *There exists a positive constant $C_{11}$, such that for any $\widetilde{w} \in \widetilde{W}_*$, $\left\| \widetilde{X}\widetilde{w} \right\| = C_{11}$.*

The proof is in Section E.4. With $H(\widetilde{w}, y) = \widetilde{A}(y)\widetilde{w} + \widetilde{b}(y)$, the verification of Assumption A5(2) and (3) is similar to Lemmas 8 and 18 and is thus omitted. Invoking Theorem 3 then yields the convergence rate of $\mathbb{E}[L(\widetilde{w}_t)]$, i.e., the convergence rate of $d(\widetilde{w}_t, \widetilde{W}_*)^2$ by the definition of $L$. The next key observation is that $d(\widetilde{w}_t, \widetilde{W}_*)^2 = (\hat{J}_t - J_\pi)^2 + d(w_t, \overline{W}_*)^2$ (Lemma 20), which completes the proof. $\qquad\square$

## 6 Experiments

We now empirically examine linear TD with linearly dependent features. Following the practice of Sutton and Barto [2018], we use diminishing learning rates $\alpha_t = \frac{\alpha}{t+t_0}$ and $\beta_t = \frac{\beta}{t+t_0}$ which closely match our Assumption LR with $\xi = 1$ and $t_0 = 10^7$. We use a variant of Boyan's chain [Boyan, 1999] with 15 states ($|\mathcal{S}| = 15$) and 5 actions ($|\mathcal{A}| = 5$) under a uniform policy $\pi(a|s) = 1/|\mathcal{A}|$, where the feature matrix $X \in \mathbb{R}^{15 \times 5}$ is designed to be of rank 3 (more details in Section F).[3] The weight convergence to a set is indeed observed. It is within expectation that different $\lambda$ requires different $\alpha, \beta$.

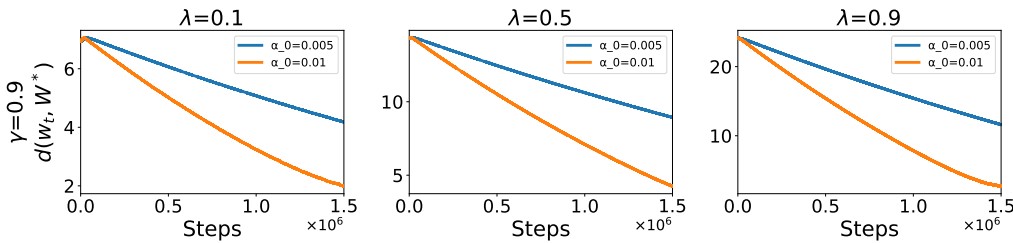

Figure 1: Convergence of (Discounted TD) with $\gamma = 0.9, \alpha_0 \in \{0.005, 0.01\}$. Curves are averaged over 10 runs with shaded regions (too small to be visible) indicating standard errors.

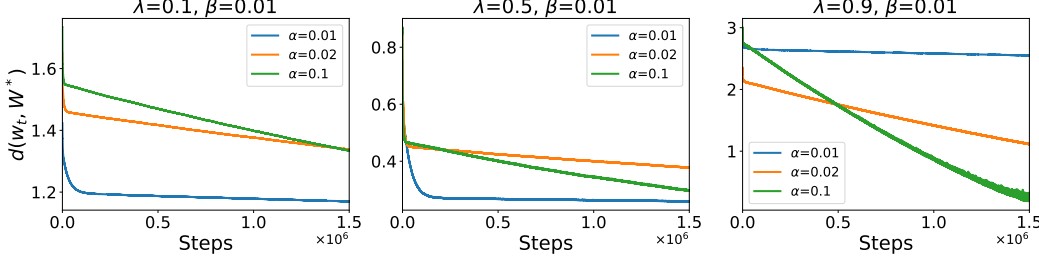

Figure 2: Convergence of (Average Reward TD) with $\beta_0 = 0.01, \alpha_0 \in \{0.01, 0.02, 0.1\}$. Curves are averaged over 10 runs with shaded regions (too small to be visible) indicating standard errors.

## 7 Conclusion

This paper provides the first finite sample analysis of linear TD with arbitrary features in both discounted and average reward settings, fulfilling the long standing desiderata of Dayan [1992], Tsitsiklis and Roy [1996, 1999], enabled by a novel stochastic approximation result concerning the convergence rate to a set. The key methodology contributions include a novel Lyapunov function based on the distance to a set and a novel decomposition of the feature matrix for the average-reward setting. We envision the techniques developed in this work can easily transfer to the analyses of other linear RL algorithms. That being said, one limitation of the work is its focus on linear function

---

[3]The code for this paper is available at `https://github.com/WennyXie/LinearTDLambda`.

approximation. Extension to neural networks with neural tangent kernels (cf. Cai et al. [2023]) is a possible future work. Another limitation is that this work considers only $L^2$ convergence rates but the convergence mode of random variables are versatile. Establishing almost sure convergence rates, $L^p$ convergence rates, and high probability concentration bounds (cf. Qian et al. [2024]) is also a possible future work. Finally, another promising direction is the integration of Polyak-Ruppert averaging (cf. Patil et al. [2023], Naskar et al. [2024]), which potentially leads to parameter-free convergence rates.

## Acknowledgments and Disclosure of Funding

This work is supported in part by the US National Science Foundation under the awards III-2128019, SLES-2331904, and CAREER-2442098, the Commonwealth Cyber Initiative's Central Virginia Node under the award VV-1Q26-001, and an Nvidia academic grant program award.

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

# A    Auxiliary Lemmas and Notations

**Lemma 12** (Discrete Gronwall Inequality, Lemma 8 in Section 11.2 of Borkar [2009]). *For non-negative real sequences $\{x_n, n \geq 0\}$ and $\{a_n, n \geq 0\}$ and scalar $L \geq 0$, it holds*

$$x_{n+1} \leq C + L \sum_{i=0}^{n} a_i x_i \quad \forall n \implies x_{n+1} \leq (C + x_0) \exp(L \sum_{i=0}^{n} a_i) \quad \forall n.$$

**Lemma 13** (Lemma 11 of Zhang et al. [2022]). *For sufficiently large $t_0$, it holds that*

$$\tau_{\alpha_t} = \mathcal{O}(\log(t + t_0)), \quad \alpha_{t - \tau_{\alpha_t}, t-1} = \mathcal{O}\left(\frac{\log(t + t_0)}{(t + t_0)^{\xi}}\right).$$

Lemma 13 ensures that there exists some $\bar{t} > 0$ (depending on $t_0$) such that for all $t \geq \bar{t}$, it holds that $t \geq \tau_{\alpha_t}$. Also, it ensures that for sufficiently large $t_0$, we have $\alpha_{t - \tau_{\alpha_t}, t-1} < 1$. Throughout the appendix, we always assume $t_0$ is sufficiently large and $t \geq \bar{t}$. We will refine (i.e., increase) $\bar{t}$ along the proof when necessary.

# B    Proofs in Section 3

## B.1    Proof of Lemma 1

*Proof.* Let $x_i \in \mathbb{R}^d$ denote the $i$-th column of $X$. Without loss of generity, let the first $m$ columns be linearly independent.
**Case 1:** When $\mathbf{1} \in \mathrm{col}(X)$, there must exist $m$ scalars $\{c_i\}$ such that $\sum_{i=1}^{m} c_i x_i = \mathbf{1}$. Apparently, at least one of $\{c_i\}$ must be nonzero. Without loss of generity, let $x_m \neq 0$. We then have

$$x_m = \frac{1}{c_m}(\mathbf{1} - \sum_{i=1}^{m-1} c_i x_i).$$

In other words, $x_m$ can be expressed as linear combination of $\{x_1, \ldots, x_{m-1}\}$ and $\mathbf{1}$. Since $X$ has a column rank $m$, we are able to express $\{x_{m+1}, \ldots, x_d\}$ by linear combination $\{x_1, \ldots, x_m\}$ and thus further by linear combination of $\{x_1, \ldots, x_{m-1}\}$ and $\mathbf{1}$. Let $Z_1 \doteq [x_1, \ldots, x_{m-1}]$ be the first $m - 1$ columns of $X$ and $Z_2 \doteq [x_m, \ldots, x_d]$ be the rest. We now know that there exists some $C \in \mathbb{R}^{(m-1) \times (d-m+1)}$ (i.e., coefficients of the lienar combination) such that

$$Z_2 = Z_1 C + [\theta_m \mathbf{1}, \ldots, \theta_d \mathbf{1}],$$

where $\theta_m, \ldots \theta_d$ are scalars (i.e., "coordinates" along the $\mathbf{1}$-axis), e.g., $\theta_m = \frac{1}{c_m}$. This means that we can express $X$ as

$$X = [Z_1 \quad Z_1 C] + [\theta_1 \mathbf{1}, \ldots, \theta_d \mathbf{1}] \tag{11}$$

with $\theta_1 = \cdots = \theta_{m-1} = 0$. Now define

$$X_1 \doteq [Z_1 \quad Z_1 C], \ X_2 \doteq [\theta_1 \mathbf{1}, \ldots, \theta_d \mathbf{1}].$$

We note that $\mathbf{1} \notin \mathrm{col}(Z_1)$. Otherwise, there would exist scalars $\{c_i'\}$ such that $\sum_{i=1}^{m-1} c_i' x_i = \mathbf{1}$. Then we get $\sum_{i=1}^{m-1}(c_i - c_i')x_i + c_m x_m = 0$, which is impossible because $\{x_i\}_{i=1,\cdots,m}$ are linearly independent. Since $\mathrm{col}(X_1) = \mathrm{col}(Z_1)$, we then have $\mathbf{1} \notin \mathrm{col}(X_1)$.
**Case 2:** When $\mathbf{1} \notin \mathrm{col}(X)$, we can trivially define $X_1 = X$ and $X_2 = 0$. Additionally, we can still further decompose $X_1$ as

$$X_1 = [Z_1 \quad Z_1 C], \tag{12}$$

where $Z_1$ is now the first $m$ columns of $X$. Apparently, we still have $\mathbf{1} \notin \mathrm{col}(X_1)$.    $\square$

**Lemma 14.** *Let Assumption 3.1 hold. Then $\overline{A} = X_1 D_\pi (P_\lambda - I) X_1, \overline{b} = X_1^\top D_\pi (r_\lambda - \frac{J_\pi}{1-\lambda}\mathbf{1})$.*

*Proof.* Apply the decomposition shown in Lemma 1, we can get

$$\overline{A} = (X_1 + X_2)^\top D_\pi (P_\lambda - I)(X_1 + X_2)$$

$$=X_1^\top D_\pi(P_\lambda - I)X_1 + X_2^\top D_\pi(P_\lambda - I)X_1 + X_1^\top D_\pi(P_\lambda - I)X_2 + X_2^\top D_\pi(P_\lambda - I)X_2$$
$$=X_1^\top D_\pi(P_\lambda - I)X_1,$$

where the last equality holds because $(P_\lambda - I)\mathbf{1} = 0$ and $\mathbf{1}^\top D_\pi(P_\lambda - I) = d_\pi^\top P_\lambda - d_\pi^\top = 0$. Similarly, for $\bar{b}$ we can obtain

$$
\begin{aligned}
\bar{b} &= (X_1 + X_2)^\top D_\pi(r_\lambda - \frac{J_\pi}{1-\lambda}\mathbf{1}) \\
&= X_1^\top D_\pi(r_\lambda - \frac{J_\pi}{1-\lambda}\mathbf{1}) + X_2^\top D_\pi(r_\lambda - \frac{J_\pi}{1-\lambda}\mathbf{1}) \\
&= X_1^\top D_\pi(r_\lambda - \frac{J_\pi}{1-\lambda}\mathbf{1}) + \theta^\top(d_\pi^\top(I - \lambda P_\pi)^{-1}r_\pi - \frac{J_\pi}{1-\lambda}) \\
&= X_1^\top D_\pi(r_\lambda - \frac{J_\pi}{1-\lambda}\mathbf{1}) + \theta^\top(\frac{1}{1-\lambda}d_\pi^\top r_\pi - \frac{J_\pi}{1-\lambda}) \\
&= X_1^\top D_\pi(r_\lambda - \frac{J_\pi}{1-\lambda}\mathbf{1}).
\end{aligned}
$$

Here, the fourth inequality holds because $d_\pi^\top(I - \lambda P_\pi) = (1-\lambda)d_\pi^\top$, which gives us $d_\pi^\top = (1-\lambda)d_\pi^\top(I - \lambda P_\pi)^{-1}$. The last inequality holds since $J_\pi = d_\pi^\top r_\pi$. This completes the proof. $\quad\square$

**Lemma 15.** *Let Assumption 3.1 hold. Then $\overline{W}_*$ is nonempty.*

*Proof.* In view of (11) and (12), we have $X_1 = [Z_1 \quad Z_1 C]$. Notably, $Z_1$ has a full column rank and $\mathbf{1} \notin \mathrm{col}(Z_1)$. Decompose $w \doteq \begin{bmatrix} w_1 \\ w_2 \end{bmatrix}$ accordingly and recall (4) and Lemma 14, we can rewrite $\overline{A}w + \bar{b} = 0$ as

$$
\begin{bmatrix} Z_1^\top \\ (Z_1 C)^\top \end{bmatrix} D_\pi(P_\lambda - I)[Z_1 \quad Z_1 C]\begin{bmatrix} w_1 \\ w_2 \end{bmatrix} = \begin{bmatrix} -Z_1^\top D_\pi(r_\lambda - \frac{J_\pi}{1-\lambda}\mathbf{1}) \\ -(Z_1 C)^\top D_\pi(r_\lambda - \frac{J_\pi}{1-\lambda}\mathbf{1}) \end{bmatrix},
$$

which thus gives us the following simultaneous equations

$$
\begin{cases}
Z_1^\top D_\pi(P_\lambda - I)Z_1 w_1 + Z_1^\top D_\pi(P_\lambda - I)Z_1 C w_2 = -Z_1^\top D_\pi(r_\lambda - \frac{J_\pi}{1-\lambda}\mathbf{1}) \\
(Z_1 C)^\top D_\pi(P_\lambda - I)Z_1 w_1 + (Z_1 C)^\top D_\pi(P_\lambda - I)Z_1 C w_2 = -(Z_1 C)^\top D_\pi(r_\lambda - \frac{J_\pi}{1-\lambda}\mathbf{1})
\end{cases}.
$$

We now prove the claim by constructing a solution. Choose any $w_2 \in \ker(Z_1 C)$ (e.g., $w_2 = 0$), the equations then become

$$
\begin{cases}
Z_1^\top D_\pi(P_\lambda - I)Z_1 w_1 = -Z_1^\top D_\pi(r_\lambda - \frac{J_\pi}{1-\lambda}\mathbf{1}) \\
C^\top Z_1^\top D_\pi(P_\lambda - I)Z_1 w_1 = -C^\top Z_1^\top D_\pi(r_\lambda - \frac{J_\pi}{1-\lambda}\mathbf{1}).
\end{cases}
$$

Since $Z_1$ is full rank and $\mathbf{1} \notin Z_1$, Lemma 7 of Tsitsiklis and Roy [1999] shows $Z_1^\top D_\pi(P_\lambda - I)Z_1$ is n.d. and thus invertible. Choose $w_1 = -(Z_1^\top D_\pi(P_\lambda - I)Z_1)^{-1}Z_1^\top D_\pi(r_\lambda - \frac{J_\pi}{1-\lambda}\mathbf{1})$ then satisfies the equations. This completes the proof.

$\quad\square$

**Lemma 16.** *Let Assumption 3.1 hold. Then*
$\overline{W}_* = \{\overline{w}_*\} + \ker(X_1)$ *and* $\ker(X_1) = \{w | Xw = c\mathbf{1}, c \in \mathbb{R}\}$.

*Proof.* For any solution $w_*, w_{**} \in \overline{W}_*$, according to the definition of $\overline{W}_*$ in (4), we have $\overline{A}w_* + \bar{b} = \mathbf{0}$ and $\overline{A}w_{**} + \bar{b} = \mathbf{0}$. That is $\overline{A}(w_* - w_{**}) = \mathbf{0}$. By multiplying $(w_* - w_{**})^\top$ on both side we can get

$$(w_* - w_{**})^\top X^\top D_\pi(P_\lambda - I)X(w_* - w_{**}) = 0.$$

According to the Perron-Frobenius theorem with Assumption 3.1, $v^\top D_\pi(P_\lambda - I)v = 0$ if and only if $v = c\mathbf{1}$ for some $c \in \mathbb{R}$. Therefore, we must have $X(w_* - w_{**}) = c\mathbf{1}$ for some $c \in \mathbb{R}$. That is, $(X_1 + X_2)(w_* - w_{**}) = c\mathbf{1}$. Recall the definition of $X_2$ in (11), we have $X_2(w_* - w_{**}) = (\theta^\top(w_* - $

$w_{**})\mathbf{1}$. This means $X_1(w_* - w_{**}) = c'\mathbf{1}$ with $c' = c - \theta^\top(w_* - w_{**})$. Since $\mathbf{1} \notin \mathrm{col}(X_1)$, we must have $c' = 0$. That is, $w_* - w_{**} \in \ker(X_1)$. Thus, we have established that $\overline{W}_* = \{\overline{w}_*\} + \ker(X_1)$.

Furthermore, if $w \in \ker(X_1)$, we have $Xw = (X_1 + X_2)w = (\theta^\top w)\mathbf{1}$. If $Xw = c\mathbf{1}$, we have $X_1 w = c\mathbf{1} - X_2 w = (c - \theta^\top w)\mathbf{1}$. But $\mathbf{1} \notin \mathrm{col}(X_1)$. So we must have $c - \theta^\top w = 0$, i.e., $w \in \ker(X_1)$. This completes the proof of $\ker(X_1) = \{w | Xw = c\mathbf{1}, c \in \mathbb{R}\}$.

$\square$

## C  Proofs in Section 5.1

**Lemma 17.** *For sufficiently large $t_0$, there exists a constant $C_{17}$ such that the following statement holds. For any $t \geq \overline{t}$ and any $i \in [t - \tau_{\alpha_t}, t]$, it holds that*
$$\left\| w_i - w_{t-\tau_{\alpha_t}} \right\| \leq C_{17}\alpha_{t-\tau_{\alpha_t}, i-1}(\|w_i - \Gamma(w_i)\| + 1).$$

*Proof.* In this proof, to simplify notations, we define shorthand $t_1 \doteq t - \tau_{\alpha_t}$ and $C_x \doteq \max_s \|x(s)\|$. Given Lemma 13, we can select a sufficiently large $t_0$ such that for any $t \geq \overline{t}$,
$$\exp\left(C_{A5}C_x\alpha_{t-\tau_{\alpha_t}t-1}\right) < 3,$$
$$C_{A5}C_x\alpha_{t-\tau_{\alpha_t}t-1} < \frac{1}{6}.$$
We then bound $\|w_i - w_{t_1}\|$ as

$$\|w_i - w_{t_1}\| \leq \sum_{k=t_1}^{i-1} \|\alpha_k H(w_k, Y_{k+1})\|$$

$$\leq \sum_{k=t_1}^{i-1} \alpha_k C_{A5}(\|Xw_k - Xw_{t_1}\| + \|Xw_{t_1}\| + 1) \quad \text{(Assumption A5)}$$

$$\leq \sum_{k=t_1}^{i-1} \alpha_k C_{A5}(\|Xw_{t_1}\| + 1) + \sum_{k=t_1}^{i-1} \alpha_k C_{A5}(\|Xw_k - Xw_{t_1}\|)$$

$$\leq \sum_{k=t_1}^{i-1} \alpha_k C_{A5}(\|Xw_{t_1}\| + 1) + \sum_{k=t_1}^{i-1} \alpha_k C_{17,1}(\|w_k - w_{t_1}\|)$$

$$\leq C_{A5}\alpha_{t_1, i-1}(\|Xw_{t_1}\| + 1)\exp(C_{17,1}\alpha_{t_1, t-1}), \quad \text{(Lemma 12)}$$

where $C_{17,1} \doteq C_{A5}C_x$. We then have
$$\|w_i - w_{t_1}\|$$
$$\leq C_{A5}\alpha_{t_1, i-1}(\|Xw_i - Xw_{t_1}\| + \|Xw_i\| + 1)\exp(C_{17,1}\alpha_{t_1, t-1})$$
$$\leq C_{A5}C_x\exp(C_{17,1}\alpha_{t_1, t-1})\alpha_{t_1, i-1}\|w_i - w_{t_1}\| + \exp(C_{17,1}\alpha_{t_1, t-1})(\|Xw_i\| + 1)C_{A5}\alpha_{t_1, i-1}$$
$$\leq \frac{1}{2}\|w_i - w_{t_1}\| + C_{17,2}\alpha_{t_1, i-1}(\|Xw_i\| + 1),$$
where $C_{17,2} \doteq 3C_{A5}$. Thus, we have
$$\|w_i - w_{t_1}\| \leq 2C_{17,2}\alpha_{t_1, i-1}(\|Xw_i\| + 1)$$
$$\leq 2C_{17,2}\alpha_{t_1, i-1}(C_2(\|w_i - \Gamma(w_i)\| + 1) + 1)$$
$$\leq C_{17}\alpha_{t_1, i-1}(\|w_i - \Gamma(w_i)\| + 1),$$
where $C_{17} \doteq 2C_{17,2}(C_2 + 1)$. This completes the proof. $\square$

### C.1  Proof of Lemma 2

*Proof.*
$$\|Xw\| = \|Xw - X\Gamma(w) + X\Gamma(w)\|$$
$$\leq \|X(w - \Gamma(w))\| + \|X\Gamma(w)\|$$
$$\leq \|X\|\|w - \Gamma(w)\| + C_{A5} \quad \text{(Assumption A5)}$$

$\square$

## C.2 Proof of Lemma 3

*Proof.* According to the definition of $H(w_t, Y_t)$ in (10),

$$
\begin{aligned}
&\|H(w_t, Y_t)\|^2 \\
&\leq C_{A5}^2(\|Xw_t\| + 1)^2 \quad \text{(By Assumption A5)} \\
&\leq 2C_{A5}^2(\|Xw_t\|^2 + 1) \\
&\leq 2C_{A5}^2(C_2^2(\|w_t - \Gamma(w_t)\| + 1)^2 + 1) \\
&\leq 2C_{A5}^2(2C_2^2(\|w_t - \Gamma(w_t)\|^2 + 1) + 1) \\
&\leq C_3(\|w_t - \Gamma(w_t)\|^2 + 1),
\end{aligned}
$$

where $C_3 \doteq 2C_{A5}^2(2C_2^2 + 1)$. This completes the proof. $\qquad\square$

## C.3 Proof of Lemma 4

*Proof.* We first decompose $\langle w_t - \Gamma(w_t), H(w_t, Y_t) - h(w_t) \rangle$ into three components similarly to Srikant and Ying [2019] as

$$
\begin{aligned}
&\langle w_t - \Gamma(w_t), H(w_t, Y_t) - h(w_t) \rangle \\
&= \underbrace{\langle (w_t - \Gamma(w_t)) - (w_{t-\tau_{\alpha_t}} - \Gamma(w_{t-\tau_{\alpha_t}})), H(w_t, Y_t) - h(w_t) \rangle}_{T_1} \\
&\quad + \underbrace{\langle w_{t-\tau_{\alpha_t}} - \Gamma(w_{t-\tau_{\alpha_t}}), H(w_t, Y_t) - H(w_{t-\tau_{\alpha_t}}, Y_t) + h(w_{t-\tau_{\alpha_t}}) - h(w_t) \rangle}_{T_2} \\
&\quad + \underbrace{\langle w_{t-\tau_{\alpha_t}} - \Gamma(w_{t-\tau_{\alpha_t}}), H(w_{t-\tau_{\alpha_t}}, Y_t) - h(w_{t-\tau_{\alpha_t}}) \rangle}_{T_3}.
\end{aligned}
$$

We leverage Lemma 2 and (9) to bound them one by one as follows.
**Bounding $T_1$:**

$$
T_1 \leq \underbrace{\left\| (w_t - \Gamma(w_t)) - (w_{t-\tau_{\alpha_t}} - \Gamma(w_{t-\tau_{\alpha_t}})) \right\|}_{T_{11}} \cdot \underbrace{\| H(w_t, Y_t) - h(w_t) \|}_{T_{12}}.
$$

For the first term, we have

$$
\begin{aligned}
T_{11} &= \left\| w_t - \Gamma(w_t) - w_{t-\tau_{\alpha_t}} - \Gamma(w_{t-\tau_{\alpha_t}}) \right\| \\
&\leq \left\| w_t - w_{t-\tau_{\alpha_t}} \right\| + \left\| \Gamma(w_t) - \Gamma(w_{t-\tau_{\alpha_t}}) \right\| \\
&\leq 2 \left\| w_t - w_{t-\tau_{\alpha_t}} \right\| \quad \text{(Since } W_* \text{ is convex)} \\
&\leq 2C_{17}\alpha_{t-\tau_{\alpha_t}, t-1}(\|w_t - \Gamma(w_t)\| + 1) \quad \text{(Lemma 17)}
\end{aligned}
$$

For the second term, we have

$$
\begin{aligned}
T_{12} &\leq C_{A5}(\|Xw_t\| + 1) + C_{A5}(\|Xw_t\| + 1) \\
&\leq 2C_{A5}(C_2(\|w_t - \Gamma(w_t)\| + 1) + 1) \\
&\leq C_{4,1}(\|w_t - \Gamma(w_t)\| + 1),
\end{aligned}
$$

where $C_{4,1} \doteq 2C_{A5}(C_2 + 1)$. Therefore, we can get

$$
T_1 \leq 2C_{17}C_{4,1}\alpha_{t-\tau_{\alpha_t}, t-1}(\|w_t - \Gamma(w_t)\| + 1)^2.
$$

Choosing $C_{4,a} \doteq 4C_{17}C_{4,1}$ then yields the bound

$$
T_1 \leq C_{4,a}\alpha_{t-\tau_{\alpha_t}, t-1}(\|w_t - \Gamma(w_t)\|^2 + 1).
$$

**Bounding $T_2$:**

$$
\begin{aligned}
T_2 &= \langle w_{t-\tau_{\alpha_t}} - \Gamma(w_{t-\tau_{\alpha_t}}), H(w_t, Y_t) - H(w_{t-\tau_{\alpha_t}}, Y_t) + h(w_{t-\tau_{\alpha_t}}) - h(w_t) \rangle \\
&\leq \underbrace{\left\| w_{t-\tau_{\alpha_t}} - \Gamma(w_{t-\tau_{\alpha_t}}) \right\|}_{T_{21}} \cdot \underbrace{\left\| H(w_t, Y_t) - H(w_{t-\tau_{\alpha_t}}, Y_t) + h(w_{t-\tau_{\alpha_t}}) - h(w_t) \right\|}_{T_{22}}.
\end{aligned}
$$

For the first term, we have:

$$
\begin{aligned}
T_{21} &= \left\| (w_{t-\tau_{\alpha_t}} - \Gamma(w_{t-\tau_{\alpha_t}})) - (\Gamma(w_t) - \Gamma(w_t)) \right\| \\
&\leq \left\| w_{t-\tau_{\alpha_t}} - \Gamma(w_t) \right\| + \left\| \Gamma(w_t) - \Gamma(w_{t-\tau_{\alpha_t}}) \right\| \\
&\leq \left\| w_{t-\tau_{\alpha_t}} - \Gamma(w_t) \right\| + \left\| w_t - w_{t-\tau_{\alpha_t}} \right\| \\
&\leq \left\| w_t - \Gamma(w_t) + w_{t-\tau_{\alpha_t}} - w_t \right\| + \left\| w_t - w_{t-\tau_{\alpha_t}} \right\| \\
&\leq \left\| w_t - \Gamma(w_t) \right\| + 2 \left\| w_t - w_{t-\tau_{\alpha_t}} \right\| \\
&\leq \left\| w_t - \Gamma(w_t) \right\| + 2C_{17}\alpha_{t-\tau_{\alpha_t},t-1}(\left\| w_t - \Gamma(w_t) \right\| + 1) \quad \text{(Lemma 17)} \\
&\leq C_{4,2}(\left\| w_t - \Gamma(w_t) \right\| + 1). \quad \text{(Lemma 13)}
\end{aligned}
\tag{13}
$$

For the second term, we have:

$$
\begin{aligned}
T_{22} &\leq \left\| H(w_t, Y_t) - H(w_{t-\tau_{\alpha_t}}, Y_t) \right\| + \left\| h(w_t) - h(w_{t-\tau_{\alpha_t}}) \right\| \\
&\leq 2C_{\text{A1}} \left\| w_{t-\tau_{\alpha_t}} - w_t \right\| \\
&\leq C_{4,3}C_{17}\alpha_{t-\tau_{\alpha_t},t-1}(\left\| w_t - \Gamma(w_t) \right\| + 1). \quad \text{(Lemma 17)}
\end{aligned}
\tag{14}
$$

Combine the result in (13) and (14), we have:

$$
T_2 \leq C_{4,2}C_{4,3}C_{17}\alpha_{t-\tau_{\alpha_t},t-1}(\left\| w_t - \Gamma(w_t) \right\| + 1)^2.
$$

Choosing $C_{4,b} \doteq 2C_{4,2}C_{4,3}C_{17}$ then yield the bound

$$
T_2 \leq C_{4,b}\alpha_{t-\tau_{\alpha_t},t-1}(\left\| w_t - \Gamma(w_t) \right\|^2 + 1).
$$

**Bounding $T_3$:**

$$
T_3 = \left\langle w_{t-\tau_{\alpha_t}} - \Gamma(w_{t-\tau_{\alpha_t}}), H(w_{t-\tau_{\alpha_t}}, Y_t) - h(w_{t-\tau_{\alpha_t}}) \right\rangle.
$$

Take expectation on both sides, we can get

$$
\begin{aligned}
\mathbb{E}[T_3] &= \mathbb{E}\left[ \left\langle w_{t-\tau_{\alpha_t}} - \Gamma(w_{t-\tau_{\alpha_t}}), H(w_{t-\tau_{\alpha_t}}, Y_t) - h(w_{t-\tau_{\alpha_t}}) \right\rangle \right] \\
&= \mathbb{E}\left[ \mathbb{E}\left[ \left\langle w_{t-\tau_{\alpha_t}} - \Gamma(w_{t-\tau_{\alpha_t}}), H(w_{t-\tau_{\alpha_t}}, Y_t) - h(w_{t-\tau_{\alpha_t}}) \right\rangle \big|_{Y_{t-\tau_{\alpha_t}}}^{w_{t-\tau_{\alpha_t}}} \right] \right] \\
&= \mathbb{E}\left[ \left\langle w_{t-\tau_{\alpha_t}} - \Gamma(w_{t-\tau_{\alpha_t}}), \mathbb{E}\left[ H(w_{t-\tau_{\alpha_t}}, Y_t) - h(w_{t-\tau_{\alpha_t}}) \big|_{Y_{t-\tau_{\alpha_t}}}^{w_{t-\tau_{\alpha_t}}} \right] \right\rangle \right] \\
&\leq \mathbb{E}\left[ \underbrace{\left\| w_{t-\tau_{\alpha_t}} - \Gamma(w_{t-\tau_{\alpha_t}}) \right\|}_{T_{31}} \cdot \underbrace{\left\| \mathbb{E}\left[ H(w_{t-\tau_{\alpha_t}}, Y_t) - h(w_{t-\tau_{\alpha_t}}) \big|_{Y_{t-\tau_{\alpha_t}}}^{w_{t-\tau_{\alpha_t}}} \right] \right\|}_{T_{32}} \right].
\end{aligned}
$$

We have

$$
\begin{aligned}
T_{32} &\leq \alpha_t(\left\| Xw_{t-\tau_{\alpha_t}} \right\| + 1) \quad \text{(By (7) and (9))} \\
&\leq \alpha_t(\left\| Xw_{t-\tau_{\alpha_t}} - Xw_t \right\| + \left\| Xw_t \right\| + 1) \\
&\leq \alpha_t(\left\| Xw_{t-\tau_{\alpha_t}} - Xw_t \right\| + C_2(\left\| w_t - \Gamma(w_t) \right\| + 1) + 1) \\
&\leq \alpha_t(C_{17}C_x\alpha_{t-\tau_{\alpha_t},t-1}(\left\| w_t - \Gamma(w_t) \right\| + 1) + C_2\left\| w_t - \Gamma(w_t) \right\| + C_2 + 1) \\
&\leq C_{4,4}\alpha_t(\left\| w_t - \Gamma(w_t) \right\| + 1).
\end{aligned}
$$

Thus, together with (13), we obtain

$$
\mathbb{E}[T_3] \leq C_{4,c}\alpha_t(\left\| w_t - \Gamma(w_t) \right\|^2 + 1),
$$

where $C_{4,c} \doteq C_{4,2}C_{4,4}$. Finally, denote $C_4 \doteq C_{4,a} + C_{4,b} + C_{4,c}$ then completes the proof. $\qquad \square$

## C.4 Proof of Lemma 5

*Proof.* We recall that

$$\|w_t - \Gamma(w_t)\|^2 = 2L(w_t).$$

Aligning Assumption A4, Lemmas 3 and 4 with (8), we get

$$\mathbb{E}[L(w_{t+1})]$$

$$\leq (1 - C_{A4}\alpha_t)\mathbb{E}[L(w_t)] + C_4\alpha_t\alpha_{t-\tau_{\alpha_t},t-1}(2\mathbb{E}[L(w_t)] + 1) + \frac{C_3}{2}\alpha_t^2 + C_3\alpha_t^2\mathbb{E}[L(w_t)]$$

$$\leq (1 - 2C_{A4}\alpha_t + 2C_4\alpha_t\alpha_{t-\tau_{\alpha_t},t-1} + C_3\alpha_t^2)\mathbb{E}[L(w_t)] + C_4\alpha_t\alpha_{t-\tau_{\alpha_t},t-1} + \frac{C_3}{2}\alpha_t^2.$$

Furthermore, we aim to derive an upper bound for $\mathbb{E}[L(w_t)]$ that depends on the initial expected loss $\mathbb{E}[L(w_0)]$ and decreases over time. First, let's denote the coefficients as $C_t$ and $D_t$:

$$C_t \doteq 1 - 2C_{A4}\alpha_t + 2C_4\alpha_t\alpha_{t-\tau_{\alpha_t},t-1} + C_3\alpha_t^2,$$

$$D_t \doteq C_4\alpha_t\alpha_{t-\tau_{\alpha_t},t-1} + \frac{C_3}{2}\alpha_t^2.$$

For sufficiently large $t_0$ and $t \geq \bar{t}$, we obtain $4C_4\alpha_{t-\tau_{\alpha_t},t-1} + C_3\alpha_t < C_{A4}$. Thus, the recursive inequality further becomes:

$$\mathbb{E}[L(w_{t+1})] \leq (1 - C_{A4}\alpha_t)\mathbb{E}[L(w_t)] + D_t,$$

where $D_t = \mathcal{O}(\alpha_t\alpha_{t-\tau_{\alpha_t},t-1})$. $\qquad\square$

## C.5 Proof of Theorem 3

*Proof.* To express $\mathbb{E}[L(w_t)]$ in terms of $\mathbb{E}[L(w_0)]$, we recursively apply the inequality:

$$\mathbb{E}[L(w_t)] \leq \prod_{i=\bar{t}}^{t}(1 - C_{A4}\alpha_i)\mathbb{E}[L(w_{\bar{t}})] + \sum_{j=\bar{t}}^{t}\left(\prod_{i=j+1}^{t}(1 - C_{A4}\alpha_i)\right)D_j.$$

Denote $E_1 \doteq \prod_{i=\bar{t}}^{t}(1 - C_{A4}\alpha_i)\mathbb{E}[L(w_{\bar{t}})]$, $E_2 \doteq \sum_{j=\bar{t}}^{t}\left(\prod_{i=j+1}^{t}(1 - C_{A4}\alpha_i)\right)\frac{\ln(j+t_0)}{(j+t_0)^{2\xi}}$, and $\kappa = C_{A4}\alpha$. Recall we have $\alpha_t = \frac{\alpha}{(t+t_0)^\xi}$. For $E_1$, set $t_0 > \kappa = C_{A4}\alpha$, we have

$$\prod_{i=\bar{t}}^{t}(1 - C_{A4}\alpha_i)\mathbb{E}[L(w_{\bar{t}})] = \prod_{i=\bar{t}}^{t}\left(1 - \frac{C_{A4}\alpha}{(i+t_0)^\xi}\right)\mathbb{E}[L(w_{\bar{t}})]$$

$$\leq \prod_{i=\bar{t}}^{t}\left(1 - \frac{\kappa}{i+t_0}\right)\mathbb{E}[L(w_{\bar{t}})]$$

$$= \mathbb{E}[L(w_{\bar{t}})]\prod_{i=\bar{t}}^{t}\frac{i+t_0-\kappa}{i+t_0}$$

$$\leq \mathbb{E}[L(w_{\bar{t}})]\left(\frac{\bar{t}+t_0}{t+t_0-\kappa}\right)^{\lfloor\kappa\rfloor}.$$

For $E_2$, we have

$$E_2 = \sum_{j=\bar{t}}^{t}\left(\prod_{i=j+1}^{t}\frac{i+t_0-\kappa}{i+t_0}\right)\frac{\ln(j+t_0)}{(j+t_0)^{2\xi}}$$

$$= \sum_{j=\bar{t}}^{t-\lfloor\kappa\rfloor}\left(\prod_{i=j+1}^{t}\frac{i+t_0-\kappa}{i+t_0}\right)\frac{\ln(j+t_0)}{(j+t_0)^{2\xi}} + \sum_{j=t-\lfloor\kappa\rfloor+1}^{t}\left(\prod_{i=j+1}^{t}\frac{i+t_0-\kappa}{i+t_0}\right)\frac{\ln(j+t_0)}{(j+t_0)^{2\xi}}$$

$$\leq \sum_{j=\bar{t}}^{t-\lfloor\kappa\rfloor}\left(\frac{j+1+t_0}{t+t_0-\kappa}\right)^{\lfloor\kappa\rfloor}\frac{\ln(j+t_0)}{(j+t_0)^{2\xi}} + \lfloor\kappa\rfloor\frac{\ln(t+t_0)}{(t-\lfloor\kappa\rfloor+1+t_0)^{2\xi}}$$

$$\leq \frac{\ln(t+t_0)}{(t+t_0-\kappa)^{\lfloor\kappa\rfloor}}C_{\text{Thm3,1}}\sum_{j=\bar{t}}^{t-\lfloor\kappa\rfloor}(j+t_0)^{\lfloor\kappa\rfloor-2\xi} + \lfloor\kappa\rfloor\frac{\ln(t+t_0)}{(t-\kappa+1+t_0)^{2\xi}}$$

**Case 1:** $\lfloor\kappa\rfloor - 2\xi > 0$

$$E_2 \leq \frac{\ln(t+t_0)}{(t+t_0-\kappa)^{\lfloor\kappa\rfloor}}C_{\text{Thm3,2}}(t-\lfloor\kappa\rfloor+t_0)^{\lfloor\kappa\rfloor-2\xi+1} + \lfloor\kappa\rfloor\frac{\ln(t+t_0)}{(t-\kappa+1+t_0)^{2\xi}}$$

$$\leq \frac{\ln(t+t_0)}{(t+t_0-\kappa)^{2\xi-1}}C_{\text{Thm3,3}} + \lfloor\kappa\rfloor\frac{\ln(t+t_0)}{(t-\kappa+1+t_0)^{2\xi}}$$

$$\leq C_{\text{Thm3,4}}\left(\frac{\ln(t+t_0)}{(t+t_0)^{2\xi-1}}\right).$$

**Case 2:** $\lfloor\kappa\rfloor - 2\xi \leq 0$

$$E_2 \leq \frac{\ln(t+t_0)}{(t+t_0-\kappa)^{\lfloor\kappa\rfloor}}C_{\text{Thm3,1}}(t-\lfloor\kappa\rfloor+1) + \lfloor\kappa\rfloor\frac{\ln(t+t_0)}{(t-\kappa+1+t_0)^{2\xi}}$$

$$\leq \frac{\ln(t+t_0)}{(t+t_0-\kappa)^{\lfloor\kappa\rfloor-1}}C_{\text{Thm3,5}} + \lfloor\kappa\rfloor\frac{\ln(t+t_0)}{(t-\kappa+1+t_0)^{2\xi}}$$

$$\leq C_{\text{Thm3,6}}\left(\frac{\ln(t+t_0)}{(t+t_0)^{\lfloor\kappa\rfloor-1}}\right).$$

Starting from the update of $w_{t+1}$, we have

$$\|w_{t+1}\| \leq \|w_t\| + \alpha_t\|H(w_t, Y_{t+1})\| \leq \|w_t\| + \alpha_t C_{\text{A1}}(\|w_t\| + 1).$$

That is, $\|w_{t+1}\| \leq \alpha_0 C_{\text{A1}} + \sum_{i=0}^{t}(\alpha_0 C_{\text{A1}} + 1)\|w_i\|$. Applying discrete Gronwall inequality, we obtain $\|w_{\bar{t}}\| \leq (C_{\text{A1}} + \|w_0\|)\exp\left(\sum_{t=0}^{\bar{t}-1}(1 + \alpha_0 C_{\text{A1}})\right) = (C_{\text{A1}} + \|w_0\|)\exp(\bar{t} + \bar{t}\alpha_0 C_{\text{A1}})$.

Denoting $C_{\text{Thm3,1}} \doteq \exp(2\bar{t} + 2\bar{t}\alpha_0 C_{\text{A1}})$ and $C_{\text{Thm3,2}} \doteq 2\max(C_{\text{Thm3,4}}, C_{\text{Thm3,6}})$ then completes the proof. $\qquad\square$

# D  Proofs in Section 5.2

## D.1  Proof of Lemma 6

*Proof.* Let $y = (s, a, s', e) \in \mathcal{Y}$ and $C_x \doteq \max_s \|x(s)\|$. We have

$$\|H(w, y) - H(w', y)\| = \left\|e(\gamma x(s')^\top - x(s)^\top)(w - w')\right\| \leq 2C_x C_e\|w - w'\|.$$

Furthermore,

$$\sup_{y\in\mathcal{Y}}\|H(0, y)\| = \sup_{y\in\mathcal{Y}}\|r(s, a)e\| \leq \max_{s,a}|r(s, a)|C_e,$$

which completes the proof. $\qquad\square$

## D.2  Proof of Lemma 18

**Lemma 18.** *There exist a constant $C_{18}$ and $\tau \in [0, 1)$ such that $\forall w$*

$$\|\mathbb{E}[H(w, Y_{t+n})\,|\,Y_t] - h(w)\| \leq C_{18}\tau^n(\|Xw\| + 1).$$

*Proof.* Given the Markov property, we only need to prove the case of $t = 1$. Recall that we use $y = (s, a, s', e)$. Define shorthand

$$\delta((s, a, s'), w) \doteq r(s, a) + \gamma x(s')^\top w - x(s)^\top w,$$

$$\delta_{n+1}(w) \doteq \delta((S_n, A_n, S_{n+1}), w).$$

By (10), we can get

$$H(w, Y_{n+1}) = \delta_{n+1}(w)e_n.$$

By expanding $e_n$, we get

$$\begin{aligned}
&\mathbb{E}[H(w, Y_{n+1}) \,|\, Y_1]\\
&=\mathbb{E}[\delta_{n+1}(w)e_n \,|\, Y_1]\\
&=\mathbb{E}\left[\delta_{n+1}(w)\sum_{k=0}^{n}(\gamma\lambda)^{n-k}x(S_k) \,|\, S_0\right].
\end{aligned}$$

Now define a two-sided Markov chain $\{\bar{S}_t, \bar{A}_t\}_{t=\ldots,-2,-1,0,1,2,\ldots}$ such that $\Pr(\bar{S}_t = s) = d_\pi(s), \Pr(\bar{A}_t = a|\bar{S}_t = s) = \pi(a|s)$, i.e., the new chain always stay in the stationary distribution of the original chain. Similarly, define

$$\bar{\delta}_{n+1}(w) \doteq \delta((\bar{S}_n, \bar{A}_n, \bar{S}_{n+1}), w).$$

We then have

$$\begin{aligned}
&\mathbb{E}\left[\delta_{n+1}(w)\sum_{k=0}^{n}(\gamma\lambda)^{n-k}x(S_k) \,|\, S_0\right]\\
&=\underbrace{\mathbb{E}\left[\bar{\delta}_{n+1}(w)\sum_{k=-\infty}^{n}(\gamma\lambda)^{n-k}x(\bar{S}_k)\right]}_{f_0(n)}\\
&\quad+\underbrace{\mathbb{E}\left[\delta_{n+1}(w)\sum_{k=0}^{n}(\gamma\lambda)^{n-k}x(S_k) \,|\, S_0\right] - \mathbb{E}\left[\bar{\delta}_{n+1}(w)\sum_{k=0}^{n}(\gamma\lambda)^{n-k}x(\bar{S}_k)\right]}_{f_1(n)}\\
&\quad-\underbrace{\mathbb{E}\left[\bar{\delta}_{n+1}(w)\sum_{k=-\infty}^{-1}(\gamma\lambda)^{n-k}x(\bar{S}_k)\right]}_{f_2(n)}.
\end{aligned}$$

In the proof of Lemma 6.7 of Bertsekas and Tsitsiklis [1996], it is proved that

$$f_0(n) = Aw + b,$$

which coincides with $h(w)$. Thus the rest of the proof is dedicated to proving that $f_1(n)$ and $f_2(n)$ decay geometrically. For $f_2(n)$, we have $\left\|\bar{\delta}_{n+1}(w)x(\bar{S}_k)\right\| \leq C_{18,1}(\|Xw\| + 1)$ for some $C_{18,1}$ (cf. (16)). We then have

$$\begin{aligned}
\|f_2(n)\| &\leq C_{18,1}(\|Xw\| + 1)\sum_{k=-\infty}^{-1}(\gamma\lambda)^{n-k}\\
&=C_{18,1}(\|Xw\| + 1)(\gamma\lambda)^n\sum_{k=1}^{\infty}(\gamma\lambda)^k.
\end{aligned}$$

For $f_1(n)$, since $\{S_t\}$ adopts geometric mixing, there exists some $\tau_1 \in [0, 1)$ and $C_{18,2} > 0$ such that

$$\sum_s \left|\Pr(S_k = s) - \Pr(\bar{S}_k = s)\right| \leq C_{18,2}\tau_1^k.$$

Then we have

$$\begin{aligned}
&\mathbb{E}[\delta_{n+1}(w)x(S_k)|S_0] - \mathbb{E}\left[\bar{\delta}_{n+1}(w)x(\bar{S}_k)\right]\\
&=\sum_s \Pr(S_k = s|S_0)x(S_k)\mathbb{E}[\delta_{n+1}(w)|S_k = s] - \sum_s d_\pi(s)x(\bar{S}_k)\mathbb{E}\left[\bar{\delta}_{n+1}(w)|\bar{S}_k = s\right].
\end{aligned}$$

Noticing that $\mathbb{E}[\delta_{n+1}(w)|S_k = s] = \mathbb{E}\big[\bar{\delta}_{n+1}(w)|\bar{S}_k = s\big]$ due to the Markov property, we obtain

$$\big\|\mathbb{E}[\delta_{n+1}(w)x(S_k)|S_0] - \mathbb{E}\big[\bar{\delta}_{n+1}(w)x(\bar{S}_k)\big]\big\| \leq C_{18,2}\tau_1^k C_{18,1}(\|Xw\| + 1).$$

This means

$$\|f_2(n)\| \leq C_{18,2}C_{18,1}(\|Xw\| + 1)\sum_{k=0}^{n}(\gamma\lambda)^{n-k}\tau_1^k.$$

Noticing that

$$\sum_{k=0}^{n}(\gamma\lambda)^{n-k}\tau_1^k \leq n\max\{\gamma\lambda, \tau_1\}^n$$

then completes the proof. $\qquad\square$

### D.3 Proof of Lemma 7

*Proof.* We start with proving $\forall w \in \ker(A)^\perp, w^\top Aw \leq -C_7\|w\|^2$. This is apparently true if $w = \mathbf{0}$. Now fix any $w \in \ker(A)^\perp$ and $w \neq \mathbf{0}$, which implies that $Aw \neq \mathbf{0}$. Now we prove by contradiction that $w^\top Aw \neq 0$. Otherwise, if $w^\top Aw = 0$, we have $w^\top X^\top D_\pi(\gamma P_\lambda - I)Xw = 0$. Since $D_\pi(\gamma P_\lambda - I)$ is n.d., we then get $Xw = \mathbf{0}$, further implying $Aw = \mathbf{0}$, which is a contradiction. We have now proved that $w^\top Aw \neq 0$. We next prove that $w^\top Aw < 0$. This is from the fact that $A$ is n.d., i.e., for $\forall z \in \mathbb{R}^d, z^\top Az \leq 0$. But $w^\top Aw \neq 0$. So we must have $w^\top Aw < 0$. Finally, we use an extreme theorem argument to complete the proof. Define $Z \doteq \{w|w \in \ker(A)^\perp, \|w\| = 1\}$. Because $z \in Z$ implies $z \in \ker(A)^\perp$ and $z \neq 0$, we have $\forall z \in Z, z^\top Az < 0$. Since $Z$ is clearly compact, the extreme value theorem confirms that the function $z \mapsto z^\top Az$ obtains its minimum value in $Z$, denoted as $-C_7 < 0$, i.e., we have

$$\forall z \in Z, z^\top Az \leq -C_7. \tag{15}$$

For any $w \in \ker(A)^\perp$ and $w \neq \mathbf{0}$, we have $\frac{w}{\|w\|} \in Z$, so $w^\top Aw \leq -C_7\|w\|^2$, which completes the proof of the first part.

We now prove that $\forall w \in \mathbb{R}^d, w - \Gamma(w) \in \ker(A)^\perp$. We recall that $\Gamma$ is the orthogonal projection to $W_* = \{w \mid Aw + b = 0\}$. Since $\Gamma$ is the orthogonal projection to $W_*$, we know $w - \Gamma(w) \in W_*^\perp$. Fix any $w_* \in W_*$ and let $z \in \ker(A)$, we then have $A(w_* + z) + b = \mathbf{0}$ so $w_* + z \in W_*$. We then have

$$\langle w - \Gamma(w), z \rangle = \langle w - \Gamma(w), w_* + z \rangle - \langle w - \Gamma(w), w_* \rangle = 0 - 0 = 0,$$

confirming that $w - \Gamma(w) \in \ker(A)^\perp$, which completes the proof.

$\qquad\square$

### D.4 Proof of Lemma 8

*Proof.* Let $y = (s, a, s', e) \in \mathcal{Y}$, since $|x(s)^\top w| \leq \max_{s \in S}|x(s)^\top w| \leq \|Xw\|$, according to (10), we have

$$\begin{aligned}
\|H(w, y)\| &= \|e(r(s, a) + \gamma x(s')^\top w - x(s)^\top w)\| &\qquad (16)\\
&\leq C_e(|r(s, a)| + \gamma|x(s')^\top w| + |x(s)^\top w|)\\
&\leq C_e(C_R + (\gamma + 1)\|Xw\|)\\
&\leq C_8(\|Xw\| + 1),
\end{aligned}$$

where $C_8 \doteq C_e(C_R + \gamma + 1)$. For $\|h(w)\|$, we have

$$\|h(w)\| = \|\mathbb{E}_{y\sim d_\mathcal{Y}}[H(w, y)]\| \leq E_{y\sim d_\mathcal{Y}}[\|H(w, y)\|] \leq C_8(\|Xw\| + 1),$$

which completes the proof. $\qquad\square$

# E Proofs in Section 5.3

## E.1 Proof of Lemma 9

*Proof.* The update to $\left\{ \hat{J}_t \right\}$ in (Average Reward TD) is

$$\hat{J}_{t+1} = \hat{J}_t + \alpha_t \left( c_\beta R_{t+1} - c_\beta \hat{J}_t \right).$$

This matches the first row of

$$\widetilde{A}(Y_t)\widetilde{w}_t + \widetilde{b}(Y_t) = \begin{bmatrix} -c_\beta & 0 \\ -\Pi e_t & \Pi e_t(x(S_{t+1})^\top - x(S_t)^\top) \end{bmatrix} \begin{bmatrix} \hat{J}_t \\ \Pi w_t \end{bmatrix} + \begin{bmatrix} c_\beta R_{t+1} \\ R_{t+1}\Pi e_t \end{bmatrix}.$$

Now consider the update for $w_t$

$$w_{t+1} = w_t + \alpha_t \left( R_{t+1} - \hat{J}_t + x(S_{t+1})^\top w_t - x(S_t)^\top w_t \right) e_t.$$

Applying the projection matrix $\Pi$ on both sides yields

$$\begin{aligned} \Pi w_{t+1} - \Pi w_t &= \alpha_t \Pi \left( \left( R_{t+1} - \hat{J}_t + x(S_{t+1})^\top w_t - x(S_t)^\top w_t \right) e_t \right) \\ &= \left( R_{t+1} - \hat{J}_t + x(S_{t+1})^\top w_t - x(S_t)^\top w_t \right) \Pi e_t \\ &= \left( R_{t+1} - \hat{J}_t + x(S_{t+1})^\top \Pi w_t - x(S_t)^\top \Pi w_t \right) \Pi e_t. \end{aligned}$$

To see the last equality, we recall Lemma 1 and recall $\Pi = X_1^\dagger X_1$. We then have

$$\begin{aligned} X\Pi w &= X_1 \Pi w + \mathbf{1}\theta^\top \Pi w \\ &= X_1 w + \mathbf{1}\theta^\top \Pi w. \end{aligned}$$

This means that

$$x(s')^\top \Pi w - x(s)^\top \Pi w = x_1(s')^\top w - x_1(s)^\top w,$$

where we use $x_1(s)$ to denote the $s$-th row of $X_1$. We also have

$$\begin{aligned} x(s')^\top w - x(s)^\top w &= (x_1(s') + \theta)^\top w - (x(s) + \theta)^\top w \\ &= x_1(s')^\top w - x_1(s)^\top w, \end{aligned}$$

which confirms the last equality and then completes the proof. $\qquad\square$

## E.2 Proof of Lemma 19

**Lemma 19.** $\widetilde{A}\Gamma(\widetilde{w}) + \widetilde{b} = \mathbf{0}$

*Proof.* According to the definition of $\Gamma(\widetilde{w})$, $\Gamma(\widetilde{w}) \in \widetilde{W}_* \doteq \left\{ \begin{bmatrix} J_\pi \\ \Pi w \end{bmatrix} \middle| w \in \overline{W}_* \right\}$. We have

$$\begin{aligned} \widetilde{A} &= \mathbb{E}_{y \sim d_{\mathcal{Y}}} \left[ \widetilde{A}(y) \right] = \mathbb{E}_{(s,a,s',e) \sim d_{\mathcal{Y}}} \begin{bmatrix} -c_\beta & \mathbf{0} \\ -\Pi e & \Pi \left( e(x(s')^\top - x(s)^\top) \right) \end{bmatrix} = \begin{bmatrix} -c_\beta & \mathbf{0} \\ -\Pi \mathbb{E}_{d_{\mathcal{Y}}}[e] & \Pi \overline{A} \end{bmatrix}, \\ \widetilde{b} &= \mathbb{E}_{y \sim d_{\mathcal{Y}}} \left[ \widetilde{b}(y) \right] = \mathbb{E}_{(s,a,s',e) \sim d_{\mathcal{Y}}} \begin{bmatrix} c_\beta r(s,a) \\ r(s,a)\Pi e \end{bmatrix} = \begin{bmatrix} c_\beta J_\pi \\ \Pi \mathbb{E}_{d_{\mathcal{Y}}}[e]J_\pi + \Pi \overline{b} \end{bmatrix}. \end{aligned} \qquad (17)$$

Therefore, for the first row of $\widetilde{A}\Gamma(\widetilde{w}) + \widetilde{b}$, we get $c_\beta(J_\pi - J_\pi) = 0$. For the second row, we can get

$$\begin{aligned} &-\Pi \mathbb{E}_{d_{\mathcal{Y}}}[e]J_\pi + \Pi \overline{A}\Pi w + \Pi \mathbb{E}_{d_{\mathcal{Y}}}[e]J_\pi + \Pi \overline{b} \\ &= \Pi(\overline{A}\Pi w + \overline{b}) \\ &= \Pi(X_1^\top D_\pi(P_\lambda - I)X_1 \Pi w + \overline{b}) \\ &= \Pi(X_1^\top D_\pi(P_\lambda - I)X_1 w + \overline{b}) \\ &= \Pi(\overline{A}w + \overline{b}) \\ &= \mathbf{0}, \end{aligned}$$

where the second equality comes with the definition of $\Pi$. This completes the proof. $\qquad\square$

### E.3 Proof of Lemma 10

*Proof.* If $z = 0$, the lemma trivially holds. So now let Let $z = \begin{bmatrix} z_1 \\ z_2 \end{bmatrix} \in \mathbb{R} \times \ker(X_1)^\perp, z \neq 0$. With (17), we have

$$\widetilde{A} = \begin{bmatrix} -c_\beta & \mathbf{0} \\ -\Pi\mathbb{E}_{(s,a,s',e)\sim d_{\mathcal{Y}}}[e] & \Pi\overline{A} \end{bmatrix} = \begin{bmatrix} -c_\beta & \mathbf{0} \\ -\Pi\mathbb{E}_{d_{\mathcal{Y}}}[e] & \Pi X_1^\top D_\pi (P_\lambda - I)X_1 \end{bmatrix} \quad \text{(Lemma 14)} \quad .$$

For simplicity, define $q \doteq \mathbb{E}_{d_{\mathcal{Y}}}[e], B \doteq X_1^\top D_\pi(P_\lambda - I)X_1$. We then have

$$z^\top \widetilde{A}z = \begin{bmatrix} z_1 & z_2^\top \end{bmatrix} \begin{bmatrix} -c_\beta z_1 \\ \Pi(-qz_1 + Bz_2) \end{bmatrix} = -c_\beta z_1^2 + z_2^\top \Pi(-qz_1 + Bz_2).$$

Recall that $\Pi = X_1^\dagger X_1$ and it is symmetric, we can get

$$z_2^\top \Pi(-qz_1 + Bz_2) = (\Pi z_2)^\top(-qz_1 + Bz_2) = z_2^\top(-qz_1 + Bz_2),$$

where the last equality holds because $z_2 \in \ker(X_1^\perp)$. Thus,

$$z^\top \widetilde{A}z = -c_\beta z_1^2 - z_2^\top qz_1 + z_2^\top Bz_2.$$

We now characterize $z_2^\top Bz_2$. Apparently, $z_2^\top Bz_2 \leq 0$ always holds because $D_\pi(P_\lambda - I)$ is n.s.d. In view of (5), the equality holds only if $X_1z_2 = c\mathbf{1}$. But $\mathbf{1} \notin \text{col}(X_1)$ and $z_2 \in \ker(X_1)^\perp$. So the equality holds only when $z_2 = 0$. Now we have proved that $\forall z_2 \in \ker(X_1)^\perp, z_2 \neq 0$, it holds that $z_2^\top Bz_2 < 0$. Using the normalization trick and the extreme value theorem again (cf. (15)), we confirm that there exists some constant $C_{10,1} > 0$ such that $\forall z_2 \in \ker(X_1)^\perp$,

$$z_2^\top Bz_2 \leq -C_{10,1}\|z_2\|^2.$$

Since $z \neq 0$, we now discuss two cases.
**Case 1:** $z_1 = 0, z_2 \neq 0$**.** In this case, we have $z^\top \widetilde{A}z = z_2^\top Bz_2 < 0$.
**Case 2:** $z_1 \neq 0$**.** In this case, we have

$$z^\top \widetilde{A}z = -c_\beta z_1^2 + z_1 z_2^\top q + z_2^\top Bz_2 \leq -c_\beta z_1^2 + |z_1|\|z_2\|\|q\| - C_{10,1}\|z_2\|^2.$$

By completing squares, it is easy to see that when $c_\beta$ is sufficiently large (depending on $\|q\|$ and $C_{10,1}$), it holds $z^\top \widetilde{A}z < 0$ because $z_1 \neq 0$.

Combining both cases, we have proved that $\forall z \in \mathbb{R} \times \ker(X_1)^\perp, z \neq \mathbf{0}$, it holds that

$$z^\top \widetilde{A}z < 0.$$

Using the normalization trick and the extreme value theorem again (cf. (15)) then completes the proof. $\qquad\square$

### E.4 Proof of Lemma 11

*Proof.* By definition, $\widetilde{W}_* = \left\{ \begin{bmatrix} J_\pi \\ \Pi w \end{bmatrix} \middle| w \in \overline{W}_* \right\}$. In view of Lemma 16, let $\overline{w}_*$ be any fixed vector in $\overline{W}_*$. Then any $\widetilde{w}_* \in \widetilde{W}_*$ can be written as

$$\widetilde{w}_* = \begin{bmatrix} J_\pi \\ \Pi(\overline{w}_* + w_{\mathbf{0}}) \end{bmatrix}$$

with some $w_{\mathbf{0}} \in \ker(X_1)$. We then have

$$\widetilde{X}\widetilde{w}_* = \begin{bmatrix} J_\pi \\ X\Pi(\overline{w}_* + w_{\mathbf{0}}) \end{bmatrix} = \begin{bmatrix} J_\pi \\ X\Pi\overline{w}_* \end{bmatrix},$$

where the last equality holds because $\Pi$ is the orthogonal projection to $\ker(X_1)^\perp$. This means that $\widetilde{X}\widetilde{w}_*$ is a constant regardless of $\widetilde{w}_*$, which completes the proof. $\qquad\square$

### E.5 Proof of Lemma 20

**Lemma 20.** $(\hat{J}_t - J_\pi)^2 + d(w_t, \overline{W}_*)^2 = d(\widetilde{w}_t, \widetilde{W}_*)^2.$

*Proof.* We recall that $\Pi$ is the orthogonal projection to $\ker(X_1)^\perp$. Let $\Pi'$ be the orthogonal projection to $\ker(X_1)$. We recall from Lemma 16 that $\overline{W}_* = \{\overline{w}_*\} + \ker(X_1)$ with $\overline{w}_*$ being any fixed point in $\overline{W}_*$. Thus for any $w_* \in \overline{W}_*$, we can write it as $\overline{w}_* + w_\mathbf{0}$ with some $w_\mathbf{0} \in \ker(X_1)$. Then for any $w \in \mathbb{R}^d$, we have

$$
\begin{aligned}
d(w, \overline{W}_*)^2 &= \inf_{w_* \in \overline{W}_*} \|w - w_*\|^2 \\
&= \inf_{w_\mathbf{0} \in \ker(X_1)} \|w - \overline{w}_* - w_\mathbf{0}\|^2 \\
&= \inf_{w_\mathbf{0} \in \ker(X_1)} \|\Pi w + \Pi' w - \Pi \overline{w}_* - \Pi' \overline{w}_* - w_\mathbf{0}\|^2 \\
&= \inf_{w_\mathbf{0} \in \ker(X_1)} \|\Pi w - \Pi \overline{w}_*\|^2 + \|\Pi' w - \Pi' \overline{w}_* - w_\mathbf{0}\|^2 \\
&= \|\Pi w - \Pi \overline{w}_*\|^2,
\end{aligned}
$$

where the last equality holds because we can select $w_\mathbf{0} = \Pi' w - \Pi' \overline{w}_*$. Define $\Pi \overline{W}_* \doteq \{\Pi w | w \in \overline{W}_*\}$. Then we have

$$
\begin{aligned}
d(\Pi w, \Pi \overline{W}_*) &= \inf_{w_* \in \overline{W}_*} \|\Pi w - \Pi w_*\| \\
&= \inf_{w_\mathbf{0} \in \ker(X_1)} \|\Pi w - \Pi(\overline{w}_* + w_\mathbf{0})\| \\
&= \|\Pi w - \Pi \overline{w}_*\|,
\end{aligned}
$$

where the last equality holds because $w_\mathbf{0} \in \ker(X_1)$ and $\Pi$ is the projection to $\ker(X_1)^\perp$ so $\Pi w_\mathbf{0} = 0$. We now have $\forall w, d(w, \overline{W}_*) = d(\Pi w, \Pi \overline{W}_*)$. Then we have

$$
\begin{aligned}
&d(\widetilde{w}_t, \widetilde{W}_*)^2 \\
=&(\hat{J}_t - J_\pi)^2 + d(\widetilde{w}_t, \Pi \overline{W}_*)^2 \\
=&(\hat{J}_t - J_\pi)^2 + d(\Pi w_t, \Pi \overline{W}_*)^2 \\
=&(\hat{J}_t - J_\pi)^2 + d(w_t, \overline{W}_*)^2,
\end{aligned}
$$

which completes the proof. $\qquad\square$

## F  Details of Experiments

We use a variant of Boyan's chain [Boyan, 1999] with 15 states $(s_0, s_1, \ldots, s_{14})$ and 5 actions $(a_0, \ldots, a_4)$. The chain has deterministic transitions. For $s_2, \ldots, s_{14}$, the action $a_0$ goes to $s_{i-1}$ and the actions $a_1$ to $a_4$ go to $s_{i-2}$; $s_1$ always transitions to $s_0$; $s_0$ transitions uniformly randomly to any state. The reward function is

$$
r(s, a) = \begin{cases} 1 & \text{if} \quad s = s_0 \\ 0 & \text{otherwise} \end{cases} \quad .
$$

We use a uniform random policy $\pi(a|s) = 0.5$. The feature matrix $X \in \mathbb{R}^{15 \times 5}$ is designed to be of rank 3.

$$
X = \begin{bmatrix}
0.07 & 0.11 & 0.18 & 0.14 & 0.61 \\
0.13 & 0.19 & 0.32 & 0.26 & 0.45 \\
0.11 & 0.17 & 0.28 & 0.22 & 0.39 \\
0.24 & 0.36 & 0.60 & 0.48 & 0.84 \\
0.18 & 0.28 & 0.46 & 0.36 & 1.00 \\
0.20 & 0.30 & 0.50 & 0.40 & 1.06 \\
0.31 & 0.47 & 0.78 & 0.62 & 1.45 \\
0.29 & 0.45 & 0.74 & 0.58 & 1.39 \\
0.42 & 0.64 & 1.06 & 0.84 & 1.84 \\
0.40 & 0.62 & 1.02 & 0.80 & 1.78 \\
0.47 & 0.73 & 1.20 & 0.94 & 2.39 \\
0.53 & 0.81 & 1.34 & 1.06 & 2.23 \\
0.58 & 0.9 & 1.48 & 1.16 & 2.78 \\
0.60 & 0.92 & 1.52 & 1.20 & 2.84 \\
0.67 & 1.03 & 1.70 & 1.34 & 3.45
\end{bmatrix}
$$

Each experiment runs for $1.5 \times 10^6$ steps, averaged over 10 runs. These experiments were conducted on a server equipped with an AMD EPYC 9534 64-Core Processor, with each run taking approximately 1 minute to complete. Memory requirements are negligible.

