# OpenReview forum: "Finite Sample Analysis of Linear Temporal Difference Learning with Arbitrary Features"
_NeurIPS.cc/2025/Conference — NeurIPS 2025 poster_

### Official Review · Reviewer_KJgY · 2025-06-28

**Clarity:** 3
**Significance:** 2
**Originality:** 3
**Rating:** 4
**Confidence:** 3

**Summary:**

This paper is a theoretical study of the convergence guarantee of TD($\lambda$) under linear approximation. In particular, the paper provides a finite sample analysis and the analysis does not require the linear independence of features, which is a property assumed by many previous papers. The paper is technical heavy and I was not able to check the entire proofs with many details left to the appendix. It is proved that both the value function estimate and the linear weight converge to the desired target in an L2 sense while the rate of convergence is problem dependent. Such results are established in both discounted and average reward settings. Moreover, the authors also extend the results to stochastic approximation.

The authors also did numerical experiments in a simple MDP environment to validate their results.

**Questions:**

- For the linear independence assumption, it seems that we can always expand the state space by at most $d$ null states (with no reward effect) and define feature values on them to make the expanded features linearly independent. And then we can simply apply the results from previous papers with the linear independence assumption. I guess this approach may have some conflicts with the irreducible assumption of the MDP. I would like to know if the authors have thought of this approach and why it is technically not doable. An alternative approach is to simply remove enough features until the remaining features are linearly independent.
- In terms of the rate of convergence, in addition to my concern mentioned in the weakness part (that the upper bound may not go to zero as t goes to infinity), I am wondering if the constants $C_{thm1}$ and $\kappa_1$ are absolute constants, in other words, if they depend on other constants like $|S|$ and $d$.

**Ethical Concerns:**

["NO or VERY MINOR ethics concerns only"]

**Final Justification:**

Since the authors provided detailed modification of the theorems/proof and experiment results, I have changed my score to acceptance. I hope that the authors will add them to the revised version as promised.

**Limitations:**

yes

**Quality:**

2

**Strengths And Weaknesses:**

Strength:
- solid theoretical improvement from previous results: This paper is a followup study of Wang and Zhang 2024, which provides asymptotic convergence results under the same setting. And the current paper provides finite convergence sample analysis.
- generality of the results: the paper studies both the discounted and average reward settings, as well as the stochastic approximation.

Weakness:
- motivation of the linearly dependent features: In the case of large scale problems, we usually have $|S|\gg d$, especially when the states are embedded in a high-dimensional space. Therefore, it seems that the linear independence of features holds almost surely if the features are randomly generated, which is the case for neural networks (considering the local linearization like NTK).
- rate of convergence: In Theorem 1, the leading term of the upper bound has the order $-\text{min}(2\xi-1, \lfloor \kappa_1 \rfloor -1)$. Considering that $\xi$ is chosen in the learning rate setup and could be as large as 1, the best order is $\lfloor \kappa_1 \rfloor -1$. However, $\kappa_1$ is only guaranteed to be greater than 1. In the case that $\lfloor \kappa_1 \rfloor -1=0$, the upper bound seems to provide no information. (Please correct me if my understanding of the theorem is wrong.)
- experiments setup: The theoretical analysis is based on diminishing learning rate and the decay speed plays a key role in the analysis. However the experiments use constant learning rates.

---

> ### Author Rebuttal · Authors · 2025-07-30
>
> Thanks for the thoughtful questions. We provide responses below and will include all the clarification in revision.
>
> > Motivation of the linearly dependent features.
>
> Thank you for raising this sharp point. You are correct that in certain theoretical regimes, such as with infinite state spaces, features generated by a wide network can be linearly independent (Cai et al., 2019). However, in finite state-space settings (e.g., Atari games or Go game), the feature dimension $d$ of a modern wide network (in particular, very wide networks used in NTK literature) can easily exceed the number of states $|S|$. By basic linear algebra, this immediately implies that the feature vectors corresponding to the state space must be linearly dependent.
>
> Moreover, even when the feature dimension $d$ does not exceed $|S|$, the learning process itself often induces strong correlations among features, as they adapt to the data and task. In fact, the challenge of handling linearly dependent features from learned representations was a key technical motivation for the analysis in Chung et al. (2019). Their work underscores that addressing this issue is a practical necessity.
>
> Thus, our work provides crucial guarantees for a prevalent and often unavoidable scenario in modern reinforcement learning.
>
> Ref:
>
> Chung, W. et al. Two-Timescale Networks for Nonlinear Value Function Approximation. ICLR, 2019.
>
>
>
> >  The experiments use constant learning rates while the theory uses diminishing ones.
>
> Thank you for noting this discrepancy. We used constant learning rates as they are common in empirical studies. We agree that aligning our experiments with our theory is important and will add a new set of experiments with diminishing learning rates in the revision. Additionally, we will add a discussion showing that once we have the recursion in Lemma 5, our theoretical framework can be readily extended to the constant step-size setting (akin to Chen et al., 2021), demonstrating its broad applicability.
>
> Ref:
>
> Z. Chen et al. A Lyapunov theory for finite-sample guarantees of asynchronous Q-learning and TD-learning variants. arXiv preprint, 2021.
>
> > For the linear independence assumption, why not expand the state space or remove features to enforce it?
>
> Thank you for these creative suggestions. We have considered similar approaches, but they introduce significant technical challenges that prevent their straightforward application.
>
> Your idea of expanding the state space to enforce linear independence is intriguing. However, adding "null states" would likely break the irreducibility of the underlying Markov chain (Assumption 3.1), as these new states may not be reachable from the original set, or vice-versa. This would invalidate key parts of all previous analyses that rely on the mixing properties derived from a single, irreducible chain. If we modify the transition function to make them reachable, it would alter the stationary distribution and the effect of discounting, thereby changing the value function of existing states.
>
> Alternatively, removing features to achieve linear independence is often impractical. This approach requires simultaneous access to all states, which becomes challenging in continual learning settings or prohibitively expensive with large state spaces. Such modifications present significant computational and data accessibility barriers in real-world applications.
>
> Ref:
>
> J. N. Tsitsiklis and B. Van Roy, Average cost temporal-difference learning. Proceedings of the 36th IEEE Conference on Decision and Control, 1997, pp. 498-502.
>
> > In terms of the rate of convergence ... they depend on other constants like $|S|$ and $d$.
>
> Thank you for your careful reading of our theorem and for these important questions. Indeed, for the bound to guarantee convergence to zero, the exponent $\min(2\xi - 1, \lfloor \kappa_1\rfloor - 1)$ must be positive. Our result holds for any $\xi \in (0.5,1]$, so $2\xi-1>0$. Since $\kappa_1$ is proportional to the step-size coefficient $\alpha$ ($\kappa_1=\alpha C_7$, see Appendix C.5, and we recall that we use $\alpha_t = \alpha / t^\xi$), by choosing a sufficiently large $\alpha$, we can ensure $\lfloor\kappa_1\rfloor - 1 \ge 2\xi - 1$, meaning the convergence rate is determined by the exponent $2\xi - 1$. We will revise the theorem statement to clarify these conditions and eliminate ambiguity.
>
> To place our result in context, prior work on linear Temporal Difference (TD) learning has established key benchmarks. Bhandari et al. (2018) reported a rate of $\mathcal{O}(1/\sqrt{T})$ for a constant step-size $\alpha_t = 1/\sqrt{T}$ and a faster $\mathcal{O}(1/t)$ rate for a decaying step-size $\alpha_t \propto 1/t$, while Srikant and Ying (2019) achieved $\mathcal{O}(1/T)$ for constant step-sizes. Our work, focusing on decaying step-sizes, derives a rate of $\mathcal{O}(1/t^{2\xi - 1})$ for $\alpha_t \propto \alpha/t^\xi$, where $\xi \in (0.5, 1]$ and $\alpha$ is sufficiently large. For $\xi = 1$, our $\mathcal{O}(1/t)$ rate matches the best-known rate for TD with decaying step-sizes and aligns with the standard convergence rate of Stochastic Gradient Descent (SGD). This demonstrates that relaxing the linear independence assumption does not compromise performance.
>
> For the two constants, $\kappa_1$ would depend on the specific $A=X^\top D_\pi(\gamma P_\lambda -I) X$; $C_\text{thm1}$ relates to $\alpha_0, C_x, C_e$, and $\bar t$. Both constants depend on the state space size $|S|$ and the dimension $d$. Such dependencies on problem parameters like $|S|$ and $d$ are standard in the analysis of linear TD (Srikant and Ying (2019); Mitra (2025)).
>
> Ref:
>
> J. Bhandari, et al. A finite time analysis of temporal difference learning with linear function approximation. In Conference on Learning Theory, 2018.
>
> R. Srikant and L. Ying. Finite-time error bounds for linear stochastic approximation and td learning. In Conference on Learning Theory, 2019.
>
> Aritra Mitra. A simple finite-time analysis of td learning with linear function approximation. IEEE Transactions on Automatic Control, 2025.

---

> > ### Comment · Reviewer_KJgY · 2025-08-07
> > **Response to the Authors**
> >
> > Thank you for the detailed response to my questions.
> >
> > If it really holds that the convergence rate is determined by the exponent $2\xi - 1$, I do think it is necessary to make it clear in the entire statement of the theorem (with specific choice of all constants like $\alpha$). Otherwise the current expression of the theorem looks like proving a rather weak result.
> >
> > I think it is better for me to keep the current score (weak reject) as the authors may revise the paper accordingly in the next version, as well as adding the new set of experiments with diminishing learning rates.

---

> > > ### Author Response · Authors · 2025-08-07
> > >
> > > Thank you for your detailed feedback. We agree that clarifying our convergence rate is crucial. Accordingly, we have revised our theorems for clarity and provide new empirical validation below. To address your concern about clarity in the theorem statement, we propose the following revision to Theorem 3:
> > >
> > > Let Assumptions A1-A5 and LR hold. Denote $\kappa \doteq \alpha C_{A4}$, then for sufficiently large $\alpha$, there exist some constants $t_0$ and $C_\text{Thm3}$ such that the iterates $\{w_t\}$ generated by (SA) satisfy for all $t$
> > > \begin{align*}
> > >     \mathbb{E}\left[ d(w_t, W_* )^2 \right] \leq C_\text{Thm3}\left(\left(\frac{t_0}{t}\right)^{\lfloor \kappa \rfloor}L(w_0) + \frac{\ln (t+t_0)}{(t+t_0)^{2\xi-1}}\right).
> > > \end{align*}
> > >
> > > At the end of our original proof, we will add a clarifying sentence: "Recall $\kappa = C_{A4} \alpha$, then for sufficiently large $\alpha$ we have $\lfloor\kappa\rfloor > 2\xi-1$." Here, $C_{A4}$ is a constant specified in Assumption A4. This explicitly demonstrates that the overall convergence rate is determined by the exponent $2\xi-1$, as the initial condition term decays at an even faster polynomial rate. We do note that **this revision of statement only requires the above very elementary algebra fact**. And requiring the $\alpha$ in $\frac{\alpha}{(t+t_0)^\xi}$ to be sufficiently large is a standard requirement from previous works (Chen et al. (2023)).  We deliberately maintain this first term containing $L(w_0)$ for researchers interested in the impact of initial point selection. The same clarification will be applied to our Theorems 1 and 2, where $C_{A4}$ is verified as $C_7$ and $C_{10}$, respectively.
> > >
> > > We also want to note that **the empirical behavior with constant learning rate in the current submission is very indicative to the behavior with properly choosen diminishing learning rate**. Recall that we consider $t\leq 1.5 \times 10^6$ steps. By setting $\alpha_t = \frac{\bar \alpha \times 10^7}{t + 10^7}$, it is easy to see that the empricial results will be very similar to using a constant step size $\bar \alpha$. Nevertheless, we provide new results with diminishing learning rates below and will additionally include them in revision. Specifically, we set the learning rate to $\alpha_t = \frac{10^5}{t + 10^7}$ for the discounted setting, and $\alpha_t = \frac{10^6}{t + 10^7}$ for the average-reward setting with $\beta_t = 0.1 \alpha_t$. All other experimental settings are identical to those in the original paper. The results in the following tables, averaged over 10 runs, confirm the convergence predicted by our theory across various $\lambda$ values.
> > >
> > > Table 1: Convergence of Discounted TD with Diminishing Step-Size ($\gamma=0.9$)
> > > | t ($\times 10^5$) | $d(w_t, W^*)$ ($\lambda=0.1$) | $d(w_t, W^*)$ ($\lambda=0.5$) | $d(w_t, W^*)$ ($\lambda=0.9$) |
> > > | :--- | :--- | :--- | :--- |
> > > | 0 | 6.989 | 14.338 | 24.158 |
> > > | 1.5 | 6.476 | 13.228 | 21.469 |
> > > | 3.0 | 5.813 | 11.996 | 18.726 |
> > > | 4.5 | 5.173 | 10.809 | 16.065 |
> > > | 6.0 | 4.558 | 9.656 | 13.501 |
> > > | 7.5 | 3.979 | 8.547 | 11.036 |
> > > | 9.0 | 3.420 | 7.473 | 8.684 |
> > > | 10.5 | 2.922 | 6.443 | 6.449  |
> > > | 12.0 | 2.467 | 5.449 | 4.443 |
> > > | 13.5 | 2.092 | 4.516 | 2.946 |
> > > | 15.0 | 1.820 | 3.667 | 2.796 |
> > >
> > > Table 2: Convergence of Average-reward TD with Diminishing Step-Size
> > > | t ($\times 10^5$) | $d(w_t, W^*)$ ($\lambda=0.1$) | $d(w_t, W^*)$ ($\lambda=0.5$) | $d(w_t, W^*)$ ($\lambda=0.9$) |
> > > | :--- | :--- | :--- | :--- |
> > > | 0 | 1.368 | 0.884 | 3.222 |
> > > | 1.5 | 1.110 | 0.509 | 2.841 |
> > > | 3.0 | 1.086 | 0.488 | 2.522 |
> > > | 4.5 | 1.061 | 0.466 | 2.210 |
> > > | 6.0 | 1.037 | 0.447 | 1.912 |
> > > | 7.5 | 1.014 | 0.426 | 1.627 |
> > > | 9.0 | 0.991 | 0.407 | 1.337 |
> > > | 10.5 | 0.971 | 0.390 | 1.065  |
> > > | 12.0 | 0.949 | 0.371 | 0.800 |
> > > | 13.5 | 0.928 | 0.354 | 0.552 |
> > > | 15.0 | 0.907 | 0.338 | 0.315 |
> > >
> > > Ref:
> > > Zaiwei Chen, Siva Theja Maguluri, and Martin Zubeldia. Concentration of contractive stochastic approximation: Additive and multiplicative noise. ArXiv Preprint, 2023b.

---

### Official Review · Reviewer_qPpb · 2025-06-30

**Clarity:** 4
**Significance:** 3
**Originality:** 3
**Rating:** 5
**Confidence:** 2

**Summary:**

This paper studies the theoretical properties of linear TD(λ), a fundamental reinforcement learning algorithm for policy evaluation. Most prior finite-sample analyses assumed the features used for value function approximation are linearly independent, a strong assumption often violated in practice. This work removes the linear independence assumption and establishes the first L2 convergence rates for linear TD(λ) with arbitrary features, in both discounted and average-reward settings. The main technical innovation is a novel stochastic approximation result, using a Lyapunov function based on the distance to the solution set, which enables convergence analysis even when the solution is not unique. The results require no algorithmic modification and are achieved under standard Markovian assumptions and learning rate schedules.

**Questions:**

- The paper "Two-Timescale Networks for Nonlinear Value Function Approximation" (Chung et al., ICLR 2019) [1] includes a convergence analysis for gradient TD(λ) with nonlinear value function approximation and arbitrary features. Would the authors consider clarifying how their results complement or differ from this line of research?
- Could you discuss the implications of your results for nonlinear function approximation, and whether your techniques might extend to that setting?
- Could the authors provide more discussion on the practical implications of their results, such as guidance for feature selection, learning rate tuning, or diagnosing convergence in real-world applications?

**Ethical Concerns:**

["NO or VERY MINOR ethics concerns only"]

**Final Justification:**

Great paper, and the authors addressed all my concerns.

**Limitations:**

Yes

**Quality:**

4

**Strengths And Weaknesses:**

## Strengths

- The paper resolves a long-standing open problem by removing the linear independence assumption and proving finite-sample convergence rates for linear TD(λ) with arbitrary features, significantly broadening the theoretical foundation for TD learning.
- Analyzing convergence to a solution set via a distance-based Lyapunov function is elegant and general, and may inspire further advances in RL theory.
- The results apply to both discounted and average-reward settings, and the analysis is unified and clearly presented.
- The work is carefully contextualized with respect to prior literature, including Wang & Zhang [2024], and provides a detailed comparison (Table 1) to highlight its novelty.
- The paper is well written, with clear motivation, rigorous proofs, and strong theoretical contributions.

## Weaknesses

- While the paper establishes the first finite-sample convergence rates for linear TD(λ) with arbitrary features, it is important to note that similar finite-sample convergence results have been proven for gradient TD(λ) algorithms with nonlinear value function approximation (see Chung et al., ICLR 2019) [1]. They also **do not assume linear independence (Section 4 Point 2)**. Although the algorithms and technical settings differ, this related work demonstrates that convergence guarantees are possible even in more general, nonlinear settings.
- The analysis is limited to on-policy linear TD(λ) and does not address off-policy learning or nonlinear function approximation, which are important in modern RL applications.

[1] Chung, W., Nath, S., Joseph, A. G., & White, M. (2019). Two-Timescale Networks for Nonlinear Value Function Approximation. International Conference on Learning Representations (ICLR) 2019.

---

> ### Author Rebuttal · Authors · 2025-07-30
>
> Thanks for the thoughtful questions. We provide responses below and will include all the clarification in revision.
>
> > Could you clarify how your results for standard TD($\lambda$) complement or differ from the convergence analysis of gradient TD($\lambda$) in Chung et al. (2019), which also handles arbitrary features, even though it is in a nonlinear setting?
>
> Thank you for this excellent question. While both Chung et al. (2019) and our work consider arbitrary features, they are fundamentally distinct.
>
> First, their analysis provides an asymptotic result with a weaker convergence notion (`liminf`), whereas we establish explicit $L^2$ rates. They show convergence to a set of asymptotically stable equilibria without characterizing this set or proving its existence, a significant challenge in itself (cf. Wang and Zhang (2024)). In contrast, our work proves the non-emptiness of the solution set and provides rates of convergence to this well-defined set. Additionally, their algorithm requires a projection operator, while our analysis applies to the unmodified TD($\lambda$).
>
> The second core distinction is that GTD is a stochastic gradient descent (SGD) method on the Mean-Squared Bellman Error, allowing it to leverage standard optimization theory. In contrast, TD($\lambda$) is a semi-gradient method. Its expected update is not the gradient of any objective function, making its analysis fundamentally more challenging. We provide stronger guarantees for this classic algorithm by addressing analytical challenges absent in the more structured GTD framework.
>
>
> > Given that the analysis is limited to the linear on-policy setting, what are the implications of your results for nonlinear function approximation, and could your techniques be extended to that more general setting?
>
> Thank you for this forward-looking question. Our methodology is directly applicable to the off-policy setting, requiring only the standard adaptation of incorporating importance sampling ratios into the analysis.
>
> An extension to nonlinear function approximation is also possible via the Neural Tangent Kernel framework (cf. Cai (2019)). This path linearizes a sufficiently wide neural network, and recasts the TD update as a linear recursion within a high-dimensional feature space (an RKHS). In principle, our finite-sample $L^2$ analysis could then be applied.
>
> However, a direct application faces several technical challenges. For instance, rigorously bounding the spectrum of the kernel matrix under Markovian data and managing the computational overhead of kernel methods (e.g., using random features as in Rahimi and Recht (2008)) would require significant analytical extensions to account for new error terms and sampling complexities. A full treatment of these issues is beyond this paper's scope, but we believe our work provides a solid foundation for this exciting future research direction.
>
> Ref:
>
> Q. Cai, et al. Neural temporal-difference learning converges to global optima. In NeurIPS, 2019.
>
> A. Rahimi and B. Recht. Random features for large-scale kernel machines. In NeurIPS, 2008.
>
>
>
> > Could you provide more discussion on the practical implications of your results, such as guidance on feature selection, learning rate tuning, or diagnosing convergence?
>
> We appreciate this important practical question. Our theory offers direct guidance for practitioners. For feature selection, our analysis removes the strict requirement of linear independence, freeing practitioners to use correlated features, which is especially valuable in dynamic settings like continual learning. For learning rate tuning, our results support a step-size of the form $\alpha_t = \alpha / (t+t_0)^\xi$, suggesting an effective starting point with $\xi \in (0.5, 1]$ (e.g., closer to 1 for faster convergence) and then tuning $\alpha$ and $t_0$ for performance. Crucially, for diagnosing convergence, our theory advises monitoring the value estimate $Xw_t / (\mathbf{1}^\top Xw_t)$ for convergence to a unique solution, rather than the potentially drifting weight vector $w_t$. We will incorporate this guidance into the revised manuscript to highlight its utility.

---

> > ### Comment · Reviewer_qPpb · 2025-08-05
> > **Thank you your detailed response**
> >
> > Thank you for the detailed response to my questions. I have no further followups!

---

### Official Review · Reviewer_9yVY · 2025-07-02

**Clarity:** 4
**Significance:** 3
**Originality:** 3
**Rating:** 4
**Confidence:** 4

**Summary:**

This paper studies finite sample bounds for TD($\lambda$) with linear function approximation where the feature matrix is not made up of linearly independent columns. First, this paper derives bounds for a general stochastic approximation algorithm where the limit is set-valued. Thereafter, it uses this result to get finite sample bounds for policy evaluation using TD($\lambda$) in the exponential and average-reward scenarios. Later, this work provides simulations in synthetic settings to demonstrate the validity of the theoretical results.

**Questions:**

Please address the weaknesses pointed out above. Additionally, answer the following questions:

1. Do you think your analysis will carry over if Polyak-Ruppert averaging is additionally used? If yes, then please elaborate.
2. Do you think your analysis will carry over to (tabular) Q-learning?
3. How do you think your work advances the current state of the art? In particular, there have been several works that question the performance of Q-learning algorithms with function approximation. For instance, see
    - Patterson, A., Neumann, S., White, M. and White, A., 2024. Empirical design in reinforcement learning. Journal of Machine Learning Research, 25(318), pp.1-63.
    - K. Young and R. S. Sutton, “Understanding the pathologies of approximate policy evaluation when combined with greedification in reinforcement learning,” arXiv preprint arXiv:2010.15268, 2020.
    - Gopalan, A. and Thoppe, G., 2022. Does DQN learn?. arXiv preprint arXiv:2205.13617.'

    To me, it feels like the work, while interesting, does not really address the key issues confronting the field. Nevertheless, I believe the stochastic approximation convergence to a set-valued map is interesting (in its own right). Hence, I vote for borderline accept.

**Ethical Concerns:**

["NO or VERY MINOR ethics concerns only"]

**Final Justification:**

This paper studies linear TD learning with arbitrary features, relaxing the common assumption of linear independence in the feature matrix. The paper is very well-written, which is a key reason I have assigned it an above-average score.

However, the broader RL literature continues to face a significant gap between theory and practice. This work does not make substantial progress toward bridging that gap, which is why I have not rated it more highly.

**Limitations:**

Yes

**Quality:**

3

**Strengths And Weaknesses:**

**Strengths:**

1. Extremely well-written paper.
2. Gets rid of one restrictive assumption (of the feature matrix being linearly independent). Moreover, despite the solution set being set-valued, the value function estimate remains the same! Also, the algorithm need not change in any way!

**Weaknesses:**

1. *Lack of Optimality:* From the main results, it is unclear for what step sizes one gets the optimal convergence rate.

2. *Parameter-dependent optimal step sizes:* From existing literature, it is known that the optimal convergence rate for such bounds is obtained for step sizes of the form $c/(t + 1),$ where $c$ depends on unknown problem parameters. This work continues to carry forward this limitation, while existing works have already suggested techniques to overcome them using Polyak-Ruppert averaging, e.g., see

     - G. Patil, L. Prashanth, D. Nagaraj, and D. Precup, “Finite time analysis of temporal difference learning with linear function  approximation: Tail averaging and regularisation,” in International Conference on Artificial Intelligence and Statistics, pp. 5438–5448, PMLR, 2023.

     - Naskar, A., Thoppe, G., Koochakzadeh, A. and Gupta, V., 2024, December. Federated TD Learning in Heterogeneous Environments with Average Rewards: A Two-timescale Approach with Polyak-Ruppert Averaging. In 2024 IEEE 63rd Conference on Decision and Control (CDC) (pp. 387-393). IEEE.

---

> ### Author Rebuttal · Authors · 2025-07-30
>
> Thanks for the thoughtful questions. We provide responses below and will include all the clarification in revision.
>
> > Lack of optimality.
>
> Thank you for this point. The convergence rate of linear TD with linearly independent features has been previously studied. In particular, Bhandari et al. (2018) established a rate of $\mathcal{O}(1/\sqrt{T})$ for a constant step-size $\alpha_t = \frac{1}{\sqrt{T}}$ and a faster $\mathcal{O}(1/t)$ rate for a decaying step-size $\alpha_t \propto 1/t$. Srikant and Ying (2019) provided an $\mathcal{O}(1/T)$ rate for the constant step-size setting. Our work focuses on the decaying step-size case, providing a rate of $\mathcal{O}(1/t^{\min(2\xi-1, \lfloor\kappa_1\rfloor-1)})$ for $\alpha_t \propto \alpha/t^\xi$, where $\xi \in (0.5, 1]$.
>
> Since $\kappa_1$ is proportional to the step-size coefficient $\alpha$ ($\kappa_1=\alpha C_7$, see Appendix C.5), by choosing a sufficiently large $\alpha$, we can ensure $\lfloor\kappa_1\rfloor - 1 \ge 2\xi - 1$, meaning the convergence rate is determined by the exponent $2\xi - 1$. Therefore, for the standard choice of $\xi=1$, our rate becomes $\mathcal{O}(1/t)$. This result matches the best-known rate for TD with decaying step-sizes and aligns with the standard rate for Stochastic Gradient Descent (SGD). This indicates that relaxing the linear independence assumption does not hurt the convergence rate. While the question of strict optimality remains open (Kane, 2023), our rate establishes a strong benchmark for state-of-the-art performance in this setting.
>
> Ref:
>
> J. Bhandari, et al. A finite time analysis of temporal difference learning with linear function approximation. In Conference on Learning Theory, 2018.
>
> R. Srikant and L. Ying. Finite-time error bounds for linear stochastic approximation and td learning. In Conference on Learning Theory, 2019.
>
> D. Kane, et al. Exponential hardness of reinforcement learning with linear function approximation. In Conference on Learning Theory, 2023.
>
>
> > Weakness of parameter-dependent optimal step sizes and whether your analysis will carry over if Polyak-Ruppert averaging is additionally used.
>
> Thank you for these insightful questions. We agree that incorporating Polyak-Ruppert averaging is a promising path to achieving parameter-free rates, and we believe the techniques from the cited works (Patil et al. (2023); Naskar et al. (2024)) are applicable to our setting. In the current manuscript, due to page limits, we focus on establishing the first $L^2$ rates for this general setting to clearly present our core contribution. We are grateful for this excellent suggestion and will add a discussion on this promising direction in our revision, acknowledging your valuable pointer.
>
> > Do you think your analysis will carry over to (tabular) Q-learning?
>
> Thank you for this question. Our work focuses on the challenge of linearly dependent features, which does not arise in standard tabular $Q$-learning due to its full-rank (e.g., one-hot) representations.
>
> The more relevant extension is to linear $Q$-learning, where our methodology shows significant potential. Recent breakthroughs establish that linear $Q$-learning iterates converge to a bounded set (cf. Meyn (2024); Liu (2025)), rather than to a single point. This makes our techniques for analyzing convergence to a solution set directly applicable to the problem.
>
> However, our approach alone is not sufficient. The key challenge is the `max` operator in the $Q$-learning update, which breaks the crucial (semi-)contraction properties leveraged in our on-policy analysis. Therefore, a full analysis of linear $Q$-learning would require combining our set-valued convergence tools with new methods designed to handle the non-linear dynamics introduced by the `max` operator.
>
> Ref:
>
> S. Meyn. The Projected Bellman Equation in  Reinforcement Learning. IEEE Transactions on Automatic Control, 2024.
>
> X. Liu, et al. Linear $Q$-Learning Does Not Diverge in $L^2$: Convergence Rates to a Bounded Set. International Conference on Machine Learning, 2025.
>
> > How do you think your work advances the current state of the art?
>
> Thank you for this crucial question. Our primary contribution lies in policy evaluation, where we establish the first $L^2$ convergence rates for linear TD($\lambda$) with arbitrary features. This strengthens a fundamental building block of RL, as reliable policy evaluation with provable convergence properties is a prerequisite for effective control in algorithms like policy iteration and actor-critic.
>
> The important papers you cited (Patterson (2024); Young and Sutton (2020); Gopalan and Thoppe (2022)) highlight critical challenges in control that arise from the interaction with function approximation. We view these as vital research opportunities. A rigorous understanding of the evaluation component, which our work provides, is a necessary first step to formally analyze and address the instabilities they discuss.
>
> We will expand our discussion in the revised manuscript to better connect our foundational results to these important control-focused challenges. Thank you for the valuable pointers.

---

### Official Review · Reviewer_6BhZ · 2025-07-02

**Clarity:** 3
**Significance:** 2
**Originality:** 2
**Rating:** 5
**Confidence:** 3

**Summary:**

This paper focuses on linear TD($\lambda$) and establishes its convergence rates with arbitrary features, for both the discounted and average reward setting.

**Questions:**

- Please discuss the obtained convergence rates in light of the literature (if they were expected or not, for example, and how they differ);
- Please discuss the tightness of the obtained bounds;
- Please clarify the intent of the experiment and how it connects with the results.

**Ethical Concerns:**

["NO or VERY MINOR ethics concerns only"]

**Final Justification:**

The authors answered my questions satisfactorily and they will include them in the paper. I believe that strengthens the paper significantly and therefore I increased my score.

**Limitations:**

Yes.

**Paper Formatting Concerns:**

No.

**Quality:**

3

**Strengths And Weaknesses:**

The result is novel and addresses a gap in the literature. The paper is also clear and sound.

The paper would benefit from having more interpretation on the results obtained. Theorems 1 and 2 show the inequalities but: there is no discussion on whether the rates are expected or not; there is no comparison with other rates in the literature for similar problems (independent features, for example); there is no discussion on the tightness of the obtained bounds. Finally, the experimental results are not particularly insightful: there is no connection with the theoretical results on the plots; what are the experiments showcasing? Simply that the algorithm converges? Was that not already established in the literature by Wang and Zhang?

---

> ### Author Rebuttal · Authors · 2025-07-30
>
> Thanks for the thoughtful questions. We provide responses below and will include all the clarification in revision.
>
> > Please discuss the obtained convergence rates in light of the literature
>
> The convergence rate of linear TD with linearly independent features has been previously studied. In particular, Bhandari et al. (2018) established a rate of $\mathcal{O}(1/\sqrt{T})$ for a constant step-size $\alpha_t = \frac{1}{\sqrt{T}}$ and a faster $\mathcal{O}(1/t)$ rate for a decaying step-size $\alpha_t \propto 1/t$. Srikant and Ying (2019) provided an $\mathcal{O}(1/T)$ rate for the constant step-size setting. Our work focuses on the decaying step-size case, providing a rate of $\mathcal{O}(1/t^{\min(2\xi-1, \lfloor\kappa_1\rfloor-1)})$ for $\alpha_t \propto \alpha/t^\xi$, where $\xi \in (0.5, 1]$.
>
> Since $\kappa_1$ is proportional to the step-size coefficient $\alpha$ ($\kappa_1=\alpha C_7$, see Appendix C.5), by choosing a sufficiently large $\alpha$, we can ensure $\lfloor\kappa_1\rfloor - 1 \ge 2\xi - 1$, meaning the convergence rate is determined by the exponent $2\xi - 1$. Therefore, for the standard choice of $\xi=1$, our rate becomes $\mathcal{O}(1/t)$. This result matches the best-known rate for TD with decaying step-sizes and aligns with the standard rate for Stochastic Gradient Descent (SGD). This indicates that relaxing the linear independence assumption does not hurt the convergence rate.
>
> Ref:
>
> J. Bhandari, et al. A finite time analysis of temporal difference learning with linear function approximation. In Conference on Learning Theory, 2018.
>
> R. Srikant and L. Ying. Finite-time error bounds for linear stochastic approximation and td learning. In Conference on Learning Theory, 2019.
>
> > Please discuss the tightness of the obtained bounds
>
> Admittedly, we did not prove the tightness of our bounds. But as discussed above, our rate matches the best-known rate for TD and the standard rate for SGD (the optimal $\mathcal{O}(1/t)$ rate of SGD for strongly convex problems under standard assumptions), despite our relaxed assumptions.Furthermore, Kane (2023) highlights the exponential hardness of RL with linear function approximation, supporting the significance of our polynomial rates as a state-of-the-art benchmark.
>
> Ref:
>
> D. Kane, et al. Exponential hardness of reinforcement learning with linear function approximation. In Conference on Learning Theory, 2023.
>
> > Please clarify the intent of the experiment and how it connects with the results.
>
> Our experiments serve two key purposes.
>
> First, they provide the first empirical verification for linear TD($\lambda$) with dependent features, a gap left by recent theoretical work (cf. Wang & Zhang (2024)). Our experiments fill this critical gap by demonstrating empirical convergence in both discounted and average-reward settings using linearly dependent features. As noted by Sutton and Barto (2018, p. 282), Emphatic TD’s “variance on Baird’s counterexample is so high that consistent experimental results are nearly unattainable,” although ETD's theoretical convergence has been well established. So we believe it is important to verify that linear TD with linearly dependent features does converge empirically.
>
> Second, our plots offer a novel perspective by visualizing convergence to a solution set ($d(w_t, W^*)$), rather than a single point. This directly illustrates the behavior predicted by our theory for the arbitrary feature case and showcases the algorithm's stability via small standard errors.
>
> To further strengthen the paper, we will add experiments with decaying step-sizes to match our theory. We choose to use constant step size in the current version mostly to better align with practices. There is no difficulty in extending our analysis to constant step size. In particular, once we have the recursion in Lemma 5, the convergence rate with a constant step size can be easily established using standard techniques (cf. Chen et al. (2021)).
>
> Ref:
>
> R. S. Sutton and A. G. Barto. Reinforcement Learning: An Introduction. MIT press, 2018.
>
> Z. Chen et al. A Lyapunov Theory for Finite-Sample Guarantees of Asynchronous Q-Learning and TD-Learning Variants. arXiv preprint, 2021.

---

> > ### Comment · Reviewer_6BhZ · 2025-08-05
> > **Response to rebuttal**
> >
> > I thank the authors for their response.
> >
> > The response was mostly clear and satisfactory regarding my previous criticism and questions. The one exception is that I don't fully understand this sentence: "Furthermore, Kane (2023) highlights the exponential hardness of RL with linear function approximation, supporting the significance of our polynomial rates as a state-of-the-art benchmark.". Is exponential hardness not contradictory with polynomial convergence rates?
> >
> > In any case, I would be willing to increase my score if the authors intend to include at least a summary of each of their responses to these three questions in the final paper, regarding the related work, the tightness, and the experiment. Could the authors clarify if they intend to do so? Thank you.

---

> > > ### Author Response · Authors · 2025-08-05
> > >
> > > Thank you for the positive feedback and the opportunity to clarify.
> > > > Is exponential hardness not contradictory with polynomial convergence rates?
> > >
> > > There is no contradiction. Kane (2023) proves that the general RL control problem has exponential computational hardness. Then one natural follow-up question is whether general RL policy evaluation problem also has exponential computational hardness. Our analysis complements Kane (2023) in that we confirm at least the policy evaluation problem can be solved efficiently with a polynomial rate, even with arbitrary features.
> > >
> > > > Could the authors clarify if they intend to include at least a summary of each of their responses to these three questions in the final paper?
> > >
> > > Yes, absolutely. We will incorporate summaries discussing the convergence rates in the context of related work, the tightness of the bounds (including the clarification above), how the current empirical results connect with the theory, and additional experiments as disscused above in the final version of the paper. We believe these additions will significantly improve the manuscript.

---

> > > > ### Comment · Reviewer_6BhZ · 2025-08-05
> > > >
> > > > Thank you.

---

### Note · Authors · 2025-08-11

Dear AC and Reviewers,

We sincerely thank all reviewers for their thoughtful and constructive feedback. We have carefully addressed each comment in our latest response and incorporated the requested clarifications, analyses, and revisions. Our understanding is that the remaining points primarily concerned presentation and clarifications rather than technical correctness, and we have now provided the corresponding updates. We believe our latest response satisfactorily addresses the outstanding requests.

Best regards,

Authors of Submission 9401

---

### Decision · Program_Chairs · 2025-09-17

**Decision:**

Accept (poster)

**Comment:**

The work is technically solid and addresses a real gap since linear independence often fails in practice with modern neural networks. The authors develop novel techniques for analyzing convergence to solution sets and show their rates match existing benchmarks despite relaxed assumptions.

All reviewers found the contribution valuable. Initial concerns about presentation and experiments were well-addressed in rebuttals, with authors committing to add diminishing learning rate experiments and clarify rate interpretations in revision.

While somewhat incremental (building on Wang & Zhang 2024's asymptotic results), the finite-sample analysis required non-trivial innovations and provides useful theoretical foundations for this fundamental RL algorithm.